# WebChoreArena: Evaluating Web Browsing Agents on Realistic Tedious Web Tasks

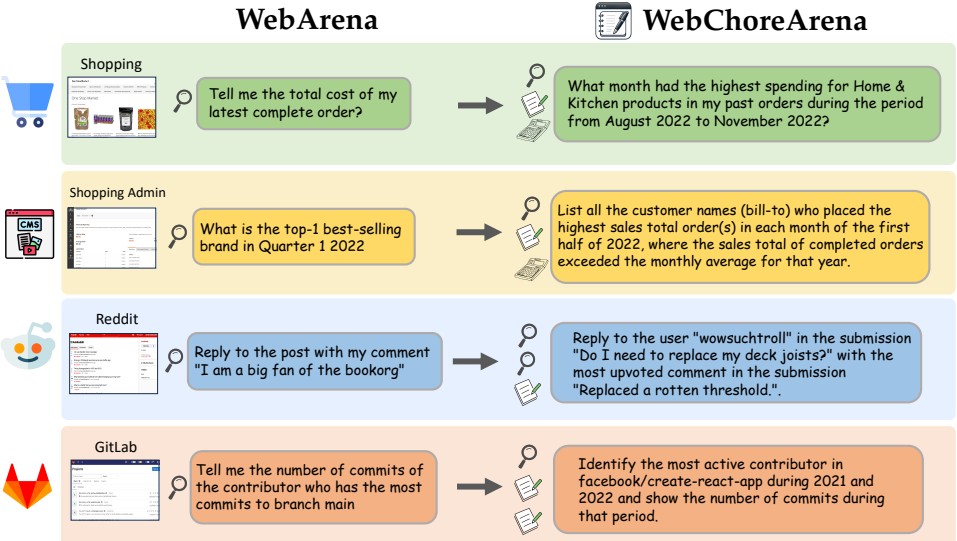

Figure 1: **The WebChoreArena challenge**. WebChoreArena introduces more complex and labor-intensive chore tasks in the web domains, pushing the boundaries of web agent capabilities. This enhanced benchmark allows for a clearer evaluation of progress in advanced models and reveals that even powerful models such as GPT-5 still have significant room for improvement.

## Abstract

Powered by a large language model (LLM), a web browsing agent operates web browsers in a human-like manner and offers a highly transparent path toward automating a wide range of everyday tasks. As web agents become increasingly capable and demonstrate proficiency in general browsing tasks, a critical question emerges: *Can they go beyond general browsing to robustly handle tasks that are tedious and complex, or chores that humans often avoid doing themselves?* In this paper, we introduce **WebChoreArena**, a new fully reproducible benchmark comprising 532 carefully curated tasks over 300+ hours, designed to address more labor-intensive and tedious tasks. WebChoreArena systematically integrates three key challenges: (i) **Massive Memory** tasks requiring accurate retrieval of large amounts of information in the observations, (ii) **Calculation** tasks demanding precise mathematical reasoning, and (iii) **Long-Term Memory** tasks necessitating long-term memory across multiple webpages. Built on top of the fully reproducible and widely adopted four WebArena environments, WebChoreArena ensures strict reproducibility and enables fair, direct comparisons with the established WebArena benchmark, offering key insights into agent progress. Our experimental results demonstrate that as LLMs evolve, significant performance improvements are observed on WebChoreArena. These findings suggest that WebChoreArena is well-suited to measure the advancement of state-of-the-art LLMs with greater clarity. Nevertheless, the results also indicate that even with GPT-5, there remains substantial room for improvement compared to WebArena, highlighting the increased challenges posed by WebChoreArena.

# 1 INTRODUCTION

Graphical User Interfaces (GUIs) serve as the primary medium through which humans perform everyday tasks. In recent years, browsing agents have gained attention as a means of automating these tasks. These agents take inputs such as accessibility trees or screenshots and produce human-like actions such as clicking and typing. While Application Programming Interfaces (APIs) and programming-based approaches enable programmatic interactions with software (Song et al., 2025), browsing agents can directly manipulate UIs, making them applicable to a wide range of web pages where APIs are unavailable. Moreover, compared to other approaches, browsing agents offer greater transparency and are more amenable to human oversight (Drouin et al., 2024). As a result, various browsing agents have been developed, continuously pushing the limits of capabilities (OpenAI, 2025a; Marreed et al., 2025; Jace AI, 2025; Yang et al., 2025; Müller & Žunič, 2024).

Among GUI agent benchmarks (Zhou et al., 2024; Koh et al., 2024; Drouin et al., 2024; He et al., 2024; Xue et al., 2025; Wei et al., 2025), WebArena (Zhou et al., 2024) has emerged as the de facto standard for evaluating web browsing agents due to its highly realistic tasks and reproducible environment (Yang et al., 2025; Wang et al., 2025b; Chae et al., 2025; Chiou et al., 2025; Qi et al., 2025; Liu et al., 2025; OpenAI, 2025a; Marreed et al., 2025; Jace AI, 2025). However, WebArena has two notable limitations. First, most of its tasks focus on general web browsing. While such tasks were effective for evaluating agent performance in earlier stages, we observe that WebArena is insufficient for precisely evaluating the performance limits and capabilities of agents with recent powerful large language models (LLMs) (refer to Figure 5). Second, approximately 20% of the tasks in WebArena contain ambiguous instructions or annotation errors (Liu et al., 2025; Yang et al., 2025; Kang, 2025) (refer to Appendix B). Although these issues had little impact when agent performance was low, they have become more serious as agents improve, and these noises limit the upper bound of performance that the benchmark can accurately capture. As LLMs and agents continue to advance, increasing task difficulty and eliminating errors in evaluation are essential for accurately assessing their capabilities for automating more tedious tasks.

In this paper, we introduce WebChoreArena, a benchmark comprising 532 human-curated tedious chore tasks that required over 300 hours of development effort. Figure 1 shows the illustration of the tasks in WebChoreArena. WebChoreArena consists of 532 human-curated tasks across the four websites used in WebArena, designed to go beyond general browsing, targeting more tedious and complex scenarios. These tasks can be broadly categorized into four types: (i) Massive Memory: tasks that require accurate memorization of a large amount of information in the observations, (ii) Calculation: tasks that involve performing mathematical reasoning based on memorized information, (iii) Long-Term Memory: tasks that require long-term memory across multiple web pages, and (iv) Others: tasks that involve special operations specific to the structure or functionality of certain websites. This benchmark enables the systematic investigation of capabilities that have been relatively underexplored in WebArena, such as memory utilization and memory-based calculation.

WebChoreArena makes the following major contributions to the research community: (1) **Proposal of Web Chore Tasks:** We introduce the notion of tedious and repetitive tasks in the web domain. Here, a chore task is defined as a series of routine operations with a clearly specified goal, but whose completion requires extensive memory, integration, and calculation of information, making them repetitive and cognitively demanding. This concept is poised to become a key focus for future research on web agents (Shao et al., 2025). (2) **Benchmark Design for In-Depth Error Analysis:** Our meticulously designed tasks can reveal critical weaknesses in each browsing agent. This enables WebChoreArena to provide precise and actionable feedback to model developers, directly contributing to the improvement of agent capabilities (Figure 4). Furthermore, by leveraging the identical simulation environment as WebArena, WebChoreArena enables rigorous comparisons of agent performance under increased task difficulty, yielding clearer insights into agent progress on more challenging tasks (Figure 5). (3) **Maximum Reproducibility and Usability for the Community:** By building upon the widely trusted, high-fidelity WebArena environment, WebChoreArena guarantees full reproducibility and ease of use for the research community.

For our experiments, to assess the progress of models, we conducted evaluations using a range of LLMs at different capability levels: GPT-4o (Hurst et al., 2024), Claude 3.7 Sonnet (Anthropic, 2024), Gemini 2.5 Pro (AI, 2025), Claude Sonnet 4 (reasoning mode) (Anthropic, 2025a), and GPT-5 (OpenAI, 2025b). We tested these LLMs with two state-of-the-art open-source web agents,

AgentOccam (Yang et al., 2025) and BrowserGym (Chezelles et al., 2024). Our key findings are summarized as follows:

- GPT-4o, a representative LLM widely used in prior WebArena research, achieved no more than 6.8% accuracy on WebChoreArena, despite reaching 42.8% on WebArena (see Table 1).
- Current powerful reasoning-based LLM agents still leave considerable room for improvement on WebChoreArena. Even GPT-5 remains below 50%, suggesting substantial room for progress (see Table 1).
- Since WebChoreArena highlights performance differences between agents more clearly than WebArena, it serves as a more reliable benchmark for evaluating the performance of increasingly powerful agents in the web domain (see Figure 5).

## 2 RELATED WORK

We focus on two key aspects: (i) the challenges of dynamic environments and reproducibility, and (ii) benchmarks targeting memory and long-term reasoning.

**Challenges of Dynamic Environments and Reproducibility.** Prior works on the real websites (Wu et al., 2025; Zhang et al., 2025b; Mialon et al., 2024) make efforts to ensure reproducibility, but their approaches have certain limitations. WebWalker (Wu et al., 2025) acknowledges on its official dataset site that the answers often change due to the dynamic nature of the webpages. Their approach relies on community-submitted pull requests to keep the dataset updated, which highlights the inherent reproducibility issues associated with dynamic content. MMInA (Zhang et al., 2025b) attempts to address the dynamics of real-world websites by evaluating not only the final output but also whether the sequence of visited URLs matches the task's intended order. However, the robustness of this method against dynamic content remains unclear due to the lack of thorough validation. GAIA (Mialon et al., 2024) tackles the reproducibility issue by fixing the time window of reference, for instance, by restricting the source content to between January and May 2018. While this is a commendable effort to improve reproducibility, it also limits the ability to design time-sensitive tasks within the benchmark. Moreover, in our settings, such as collecting large amounts of information from web pages, ensuring reproducibility becomes even more difficult than in the navigation-focused tasks these existing works typically address. Therefore, constructing WebChoreArena directly on real websites would be highly challenging.

**Benchmarks Focusing on Memory and Long-term Reasoning.** OSWorld (Xie et al., 2024), TheAgentCompany (Xu et al., 2024), WorkArena (Drouin et al., 2024), and WorkArena++ (Boisvert et al., 2024) are representative benchmarks that incorporate aspects of long-term reasoning and memory, similar to WebChoreArena. Here, we describe the clear difference from WebChoreArena. OSWorld (Xie et al., 2024) focuses on operating systems. It evaluates application manipulation tasks assuming various operating systems. While such application-level tasks are certainly important, it does not cover the complex, tedious, and repetitive tasks that we target in the web domain. TheAgentCompany (Xu et al., 2024) focuses on office tasks. It simulates the workflows of an actual software company and evaluates how well agents can autonomously complete office tasks. While it covers tasks such as project management, software engineering, and communication with simulated colleagues, it does not capture the capabilities we aim to evaluate. WorkArena (Drouin et al., 2024) and WorkArena++ (Boisvert et al., 2024) are specialized for ServiceNow, a specific enterprise platform, and do not cover general-purpose web tasks. Therefore, benchmarks that enable the evaluation of agents in more general web environments remain highly important.

Furthermore, WebChoreArena's unique strength lies in its full compatibility with WebArena, enabling the seamless transfer of the community's collective efforts for WebArena and thus accelerating advancements in this research area.

## 3 WEBCHOREARENA BENCHMARK

### 3.1 RATIONALE FOR BUILDING ON WEBARENA ENVIRONMENTS.

The reason WebChoreArena is built on the simulation environments of WebArena, rather than real-world websites, is to accurately measure agents' progress. While general navigation tasks need to

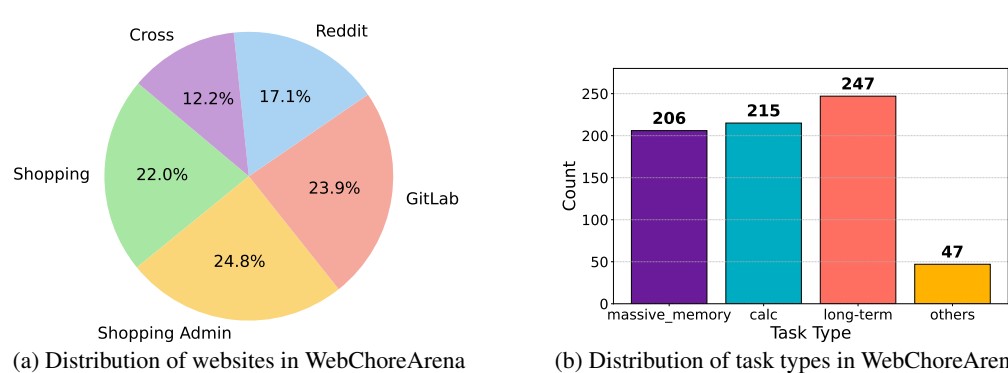

(a) Distribution of websites in WebChoreArena  (b) Distribution of task types in WebChoreArena

Figure 2: Distribution of websites and task types in WebChoreArena.

be evaluated in real-world environments (He et al., 2024; Xue et al., 2025; Yoran et al., 2024; Pan et al., 2024), the types of tasks covered by WebChoreArena, such as collecting large amounts of information from web pages, are fundamentally different from navigation-focused tasks. These tasks are particularly difficult to evaluate in real-world websites, since content such as product listings and review counts changes frequently. While establishing a scalable evaluation protocol in such dynamic settings is essential, it remains a significant challenge for the research community. Therefore, we constructed WebChoreArena on the well-established WebArena environment, which closely mirrors real-world settings, to guarantee reproducibility.

## 3.2 Overview of the WebChoreArena

Our WebChoreArena benchmark consists of 532 newly and carefully human-curated tasks. It follows the design principles of WebArena and includes four simulated websites, Shopping (e-commerce platforms), Shopping Admin (content management systems), Reddit (social forum platforms), and GitLab (collaborative development platforms), providing a fully reproducible evaluation environment. The distribution of tasks across websites is illustrated in Figure 2a: 117 tasks for Shopping, 132 for Shopping Admin, 91 for Reddit, 127 for GitLab, and 65 Cross-site tasks that require navigation across multiple websites.

## 3.3 Statistics of the WebChoreArena

WebChoreArena further categorizes the task types into the following four types:

**1. Massive Memory**: Tasks that require the agent to store a large amount of observations in memory. In the top-left of Figure 3, the agent must accurately collect review scores from a category page. These tasks evaluate the agent's ability to extract and retain all necessary information from a webpage.

**2. Calculation**: Tasks that require mathematical reasoning based on previously observed content. As illustrated in the bottom-left of Figure 3, an agent must track and sum the number of comments across the top 40 posts. This category assesses whether the agent can perform arithmetic or logical operations over stored memories from earlier steps.

**3. Long-Term Memory**: Tasks that necessitate long-term memory and reasoning across multiple web pages. For instance, in the top-right of Figure 3, the agent must first retrieve pricing rules from one page and then apply them while interacting with an order page. These tasks evaluate the agent's capacity to recall and correctly use earlier information after multiple navigational steps.

**4. Others**: Tasks involving uncommon or specialized operations, such as assigning labels in GitLab, as shown in the bottom-right of Figure 3. These problems test the agent's ability to handle unusual UI elements or actions not commonly encountered in simpler browsing scenarios.

These questions can have multiple types. For each question, we defined up to two types: a main type ('type_main') and a sub-type ('type_sub'). Figure 2b shows the distribution of the number of task types, considering both 'type_main' and 'type_sub'. More detailed information is provided in Appendix C.

Furthermore, WebChoreArena contains 451 tasks solvable with any observation, 69 tasks that require accessibility trees, and 12 tasks that require screenshots. Examples of each type are provided in Appendix C. We retained this diversity as it reflects realistic challenges encountered on actual websites. For each task, we provide only the accessibility tree as input when the task is solvable with any modality or with text alone, and we provide both the accessibility tree and a screenshot when the task requires visual information. (A more detailed explanation is given in Table A.)

### 3.4 DATASET CONSTRUCTION PIPELINE

We assigned three annotators (selected from the authors) to each of the four simulated websites. To ensure consistency in task quality across different websites, one annotator was assigned to all four websites. In total, ten annotators were involved in the task creation process. Following the creation process of WebArena, our annotators were guided to explore the websites to familiarize themselves with the websites' content and functionalities. Next, we instructed the annotators to formulate intents based on the following criteria:

**1. Emphasis on Memory-intensive Analytical Tasks.** We deliberately focused on collecting tasks that require memory, that is, tasks in which information from past observations is essential to reach the correct answer. Such tasks are common in real-world scenarios but remain largely underrepresented in WebArena. To avoid overly simplistic tasks, we first prototyped early task ideas and evaluated them using a Claude-based agent to identify model limitations and refine the task designs. This process ensured that our final tasks were both meaningful and appropriately challenging. As a result of this construction process, our tasks fall naturally into the four task categories in Sec. 3.3.

**2. Reducing Ambiguity in Task Specification and Evaluation.** We explicitly instructed annotators to eliminate ambiguities in both tasks and evaluation criteria. While handling ambiguous instructions is important for agents aiming to operate flexibly in real-world human interactions, we prioritize clear evaluability, since reliable evaluation is essential for measuring progress. In WebArena, vague instructions often lead to scenarios where agents produce reasonable answers that are incorrectly marked as failures. For example, consider the task: "*Buy the highest rated product from the category within a budget under $20.*" with the ground-truth answer: "*The website does not support sorting by rating and there are too many products in this category.*" Although the agent actually purchased a product with a 100% rating, its behavior is considered incorrect according to the ground truth. In addition, we observed that the evaluation protocol in WebArena can fail to reliably assess answers due to vague output format expectations. To mitigate ambiguity in answer evaluation, we standardized the required output formats, *e.g.,* "*Provide only the answer without any additional words.*" when aiming for exact matching with the ground truth (refer to Sec. 3.5 for evaluation details).

**3. Template-based Task Construction and Extension.** Following WebArena, we instructed annotators to create task templates and extend them to several task instances. The annotators were also responsible for developing several instantiations for each variable. This templated design enables a more robust and systematic evaluation of agent performance across tasks that share semantic similarity but exhibit diverse execution traces (Zhou et al., 2024; Koh et al., 2024).

We created a total of 117 task templates: 25 for Shopping, 29 for Shopping Admin, 20 for Reddit, 28 for GitLab, and 15 for Cross-site tasks. On average, each template yielded approximately 4.5 task instances. Here, WebArena includes several tasks based on the map website (OpenStreetMap). Although we attempted to create tasks for the map website as well, we encountered two major issues. Therefore, we decided to focus on the main four websites, aiming to build a more reliable and accessible benchmark. Details are provided in Appendix C.

To ensure the quality and correctness of each task, we conducted cross-checking with three annotators per website. Since many ambiguities were only revealed during actual task execution, we iterated through multiple rounds of inference, error analysis, and revision. This annotation process was both meticulous and labor-intensive, totaling over 300 hours of careful refinement.

### 3.5 EVALUATION PROTOCOL

We follow the established evaluation protocol of WebArena, which has been widely adopted by the community, to ensure accurate evaluation. We adopt three evaluation metrics: `string_match` for assessing textual outputs, `url_match` for verifying the final displayed URL against the ground

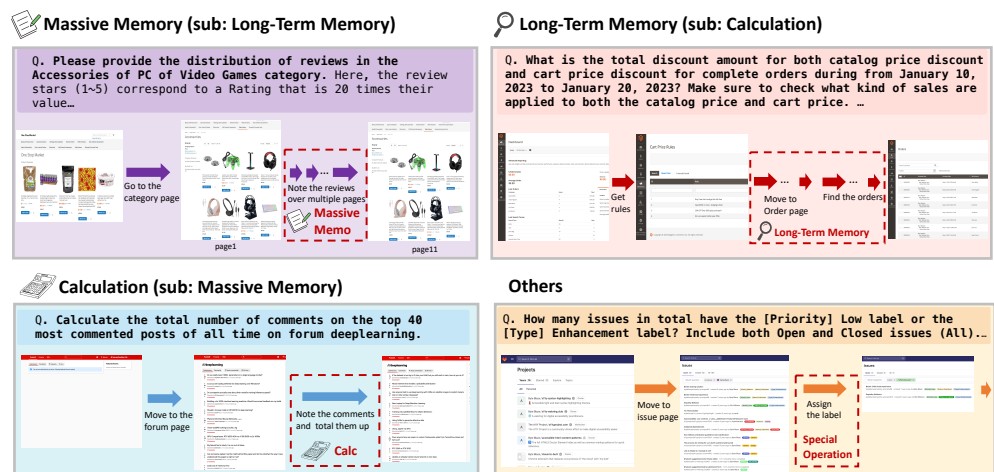

Figure 3: **Examples in each task type in WebChoreArena.** (i) Massive Memory tasks require accurately memorizing a large amount of information from the given page. (ii) Calculation tasks involve performing arithmetic operations. (iii) Long-Term Memory tasks require the agent to retain relevant information across many steps and interactions. (iv) Others involve tasks that require special or domain-specific operations.

truth, and `program_html` for functional evaluation of web interactions. The descriptions of the string-based and functional interaction evaluations are provided below.

**Evaluation of Textual Outputs (string_match).** String evaluation can be divided into the following three categories. (i) `exact_match`: A success is recorded only if the output exactly matches the ground truth. (ii) `must_include`: A success is recorded if the ground truth is included anywhere within the output. (iii) `fuzzy_match`: This function leverages a language model (GPT-4o in our implementation) to assess whether the output is semantically equivalent to the ground truth.

**Evaluation of Web Interactions (program_html).** This verifies whether the expected state change has occurred on the webpage after the agent's actions. Specifically, we extract information from designated elements on the post-action webpage using locators, and compare it against the ground truth, which determines correctness in a functional manner.

## 4 WEB BROWSING AGENTS

### 4.1 PROBLEM FORMULATION

The environment and agent can be modeled as a partially observable Markov decision process (POMDP): $\mathcal{E} = (S, A, \Omega, T, \mathcal{M})$, where $S$ represents the set of states, $A$ represents the set of actions, $\Omega$ represents the set of observations and $\mathcal{M}$ is the set of memory states. The transition function is defined as $T : S \times A \to S$, with deterministic transitions between states conditioned on actions. At each time step $t$, the environment is in some state $s_t$ (*e.g.,* a particular page), with a partial observation $o_t \in \Omega$ along with a memory buffer $M_t \in \mathcal{M}$ that stores important information from previous steps up to $t - 1$. An agent then issues an action $a_t \in A$ conditioned on both $o_t$ and the stored memory $M_t$, which results in a new state $s_{t+1} \in S$ and a new partial observation $o_{t+1} \in \Omega$ of the resulting page. Simultaneously, relevant information from $o_t$ is written to the memory, updating it to $M_{t+1}$. The action $a_t$ may be an interaction executed on the webpage or simply a string output.

### 4.2 BASELINE AGENTS

For our experiments, we referred to the WebArena leaderboard (Webarena Team) and adopted two open-source agents: a BrowserGym-based agent (Chezelles et al., 2024) and AgentOccam (Yang et al., 2025), which currently achieves state-of-the-art performance among open-source agents on WebArena. The two agents, BrowserGym and AgentOccam, are among the highest performing

Table 1: **Overall and per-website accuracy (%).** We observe that current agents significantly struggle with our WebChoreArena. Even with the most recent GPT-5–based AI agents, performance remains below 50%, indicating substantial room for improvement.

| Agent | Model | Shopping | Admin | Reddit | GitLab | Cross | Overall |
|---|---|---|---|---|---|---|---|
| - | Human | 64.0 | 46.7 | 45.0 | 41.4 | 57.1 | 50.2 |
| AgentOccam | GPT-4o | 10.3 | 4.5 | 9.9 | 7.1 | 0.0 | 6.8 |
| | Claude 3.7 Sonnet | 27.4 | 28.8 | 23.1 | 22.8 | 7.7 | 23.5 |
| | Gemini 2.5 Pro | 33.3 | 43.2 | 40.7 | 41.7 | 12.3 | 36.5 |
| | Claude Sonnet 4 | 34.2 | 37.4 | 30.8 | 36.2 | 3.1 | 31.1 |
| | GPT-5 | 56.4 | 43.9 | 53.8 | 41.7 | 21.5 | 45.1 |
| BrowserGym | GPT-4o | 4.3 | 2.3 | 5.5 | 3.9 | 0.0 | 3.4 |
| | Claude 3.7 Sonnet | 16.2 | 26.5 | 18.7 | 25.2 | 30.8 | 23.1 |
| | Gemini 2.5 Pro | 31.6 | 43.9 | 34.1 | 37.0 | 35.4 | 36.8 |
| | Claude Sonnet 4 | 35.9 | 44.7 | 25.3 | 39.4 | 38.5 | 37.4 |
| | GPT-5 | 43.6 | 61.4 | 44.0 | 48.8 | 35.4 | 48.3 |

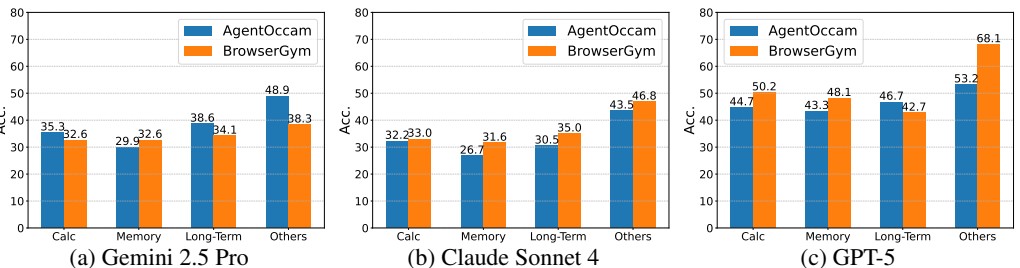

(a) Gemini 2.5 Pro  (b) Claude Sonnet 4  (c) GPT-5

Figure 4: **Comparison across different task types.** This result reveals that the methodology of the agent itself has a substantial impact on its effectiveness across different task types.

open-source agents on WebArena and serve as scalable foundations for subsequent research (Wang et al., 2025b; Zhang et al., 2025a). Evaluating them provides practical and reproducible baselines for the open-source research community. The more detailed information on these agents and the discussions on other agents' attempts are included in Appendix E.1.3.

## 5 EXPERIMENT

### 5.1 MAIN RESULTS

For our experiments, we conducted evaluations using a range of LLMs at different capability levels: GPT-4o (Hurst et al., 2024), Claude 3.7 Sonnet (Anthropic, 2024), Gemini 2.5 Pro (AI, 2025), Claude Sonnet 4 (reasoning mode) (Anthropic, 2025a), and GPT-5 (OpenAI, 2025b).

Table 1 presents the results of WebChoreArena. Figure 5 presents the comparison of the performance in WebChoreArena with that in WebArena.

**Human Performance.** We conducted human evaluations after obtaining approval from our institution's Institutional Review Board (IRB). Following the WebArena human experiment protocol, we recruited eight participants, including both undergraduate and graduate students, and had them solve one task per template. Human participants achieved higher accuracy than the current AI agents. However, their overall accuracy remains around 50%, indicating that the tasks are still quite challenging. Our error analysis suggests that these mistakes were not due to poorly designed or unsolvable questions. Instead, most errors stemmed from the complexity and tediousness of the tasks, leading to careless mistakes such as overlooking instructions or miscounting. WebChoreArena is intentionally designed to include tasks that are cognitively demanding and perceived as tedious by humans. These findings support the benchmark's goals and show that even humans have difficulty maintaining focus throughout the tasks.

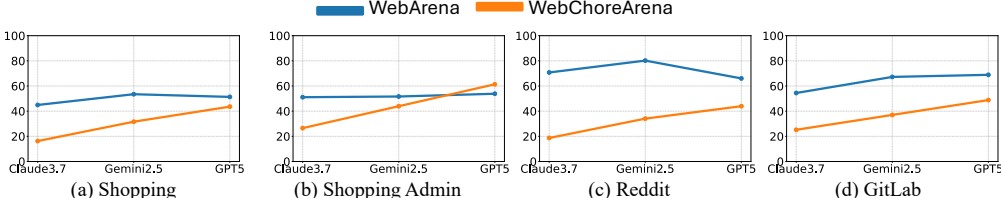

Figure 5: **Performance trend over model progress on WebArena and WebChoreArena.** The agent is BrowserGym. The results show that, in WebArena, the score does not increase even when model capabilities improve, whereas in WebChoreArena, models with higher capabilities are directly reflected in higher scores.

Table 2: **Performance with different modalities.** We evaluated three LLMs using BrowserGym across different input modalities: accessibility trees (A11y Tree) alone or with screenshots. Incorporating image inputs does not necessarily lead to overall performance improvements.

|  | Input | Shopping (#25) | Admin (#29) | Reddit (#20) | GitLab (#28) | Overall (#102) |
|---|---|---|---|---|---|---|
| Gemini2.5 | Image + A11y Tree | 24.0 | 41.4 | 20.0 | 42.9 | 33.3 |
|  | A11y Tree | 28.0 | 48.3 | 30.0 | 39.3 | 37.3 |
| Claude4.0 | Image + A11y Tree | 28.0 | 41.4 | 15.0 | 28.6 | 29.4 |
|  | A11y Tree | 40.0 | 51.7 | 15.0 | 42.9 | 39.2 |
| GPT-5 | Image + A11y Tree | 52.0 | 58.6 | 35.0 | 48.0 | 49.5 |
|  | A11y Tree | 44.0 | 58.6 | 40.0 | 53.6 | 50.0 |

Next, our main findings are as follows:

**F1: Current LLM Agents Struggles Significantly on WebChoreArena.** From Table 1, it is evident that current LLM agents struggle substantially on WebChoreArena. GPT-4o, which is widely used for WebArena evaluation and achieved 42.8% in our preliminary WebArena experiments, shows a sharp performance drop to 6.8% and 3.4% on WebChoreArena. Furthermore, even the latest powerful reasoning-based LLMs, such as GPT-5, remain below 50%, indicating considerable room for improvement. These results demonstrate that WebChoreArena poses significant challenges and highlight the necessity of more advanced models to address these tasks effectively.

**F2: WebChoreArena Enables a Clearer and Deeper Measurement of the Performance Differences among the Models.** WebChoreArena serves as a more effective benchmark for distinguishing model performance. Figure 5 illustrates the performance trends of Claude 3.7 Sonnet, Gemini 2.5 Pro, and GPT-5 on both WebArena and WebChoreArena. For WebArena (blue), performance differences remain minimal even as models advance, largely due to its dependence on task difficulty and ambiguity (Liu et al., 2025; Yang et al., 2025; Kang, 2025) (refer to Appendix Appendix B). In contrast, WebChoreArena more accurately captures the progress of increasingly capable models, thereby providing a reliable benchmark. Consequently, WebChoreArena offers model developers and evaluators clear insights into the precise progress of each agent.

**F3: WebChoreArena Enables Fine-grained Analysis of Task-specific Performance.** Figure 4 presents a detailed analysis of each agent's performance across diverse task typologies. The results underscore the significant influence of agent architecture, beyond the type of LLMs, on type-wise performance. Notably, BrowserGym performs better on Massive Memory tasks than AgentOccam. This difference can be attributed to fundamental differences in their memory management strategies: BrowserGym explicitly outputs memory at each step, while AgentOccam relies on summaries or must issue a `note [content]` action to retain important information (Further elaboration in Appendix E). In this way, analyzing the performance in each task type allows model and agent developers to receive feedback on which mechanisms should be improved.

## 5.2 ANALYSIS

**Effect on Input Modality.** We investigate the impact of input data modality on agent performance. The main experiments primarily utilized text-based inputs (*i.e.,* accessibility trees) to mitigate visual hallucinations (with the exception of only three templates requiring image inputs) following previous work (Yang et al., 2025; Zhou et al., 2024; Chiou et al., 2025). Nevertheless, analyzing how performance changes when image input (*i.e.,* screenshots) is incorporated would provide insights. We selected one task from each task template across the four websites, creating a small-set. We adopt BrowserGym for this experiment. The results are summarized in Table 2. These results indicate an overall trend of decreased performance when incorporating image inputs due to visual hallucinations (Chiou et al., 2025). However, recent GPT-5 is less affected by visual hallucinations, achieving consistently high performance regardless of the input modality.

## 6 ERROR PATTERN ANALYSIS

This section presents an analysis of the error tendencies of GPT-5 (BrowserGym). We carefully examined the failure cases and identified several distinct types of mistakes. More detailed examples of failed cases of GPT-5 are provided in Appendix C.3.

**Counting Errors.** In tasks including counting such as "XX among the top 30 posts in the forum," the model often failed to count the number of the posts correctly, repeatedly pressing the "More" button instead. Similar issues were observed across multiple websites. As another example, in tasks requiring the model to count the number of commits after a certain date, it frequently made counting mistakes. This is a failure under Massive Memory tasks. This error suggests agents require more capable memory modules or mechanisms for structured aggregation across long spans of interactions.

**Navigation Errors.** The agent often answers without sufficient exploration. In tasks requiring the retrieval of products that meet certain conditions, it would output an answer while leaving some pages unexplored. In other cases, it wasted steps by repeatedly viewing the same page. This is a failure under Long-term Memory tasks. This reveals the need to incorporate more effective planning.

**Calculation Errors.** Compared to other models, GPT-5 makes remarkably few calculation mistakes. In one task that required computing the average price of products across more than 100 categories, the model erred only in the second decimal place of the final result. Gemini 2.5 Pro started to make calculation mistakes noticeably more often when it had to add or multiply more than fifteen numbers. Given that, GPT-5 demonstrates strong calculation ability.

**Operational Errors.** We also observed several operational errors. In tasks requiring sorting by date, the agent often forgot to apply the date-based sorting. Similarly, in tasks requiring filtering by price, it neglected to apply the filter. In cases where posting to a forum was required, the agent occasionally failed to complete the posting step. This is a failure under Others task. This highlights the importance of enhancing the agent's execution strategies.

## 7 CONCLUSION AND LIMITATIONS

This paper introduces WebChoreArena, a new fully reproducible benchmark comprising 532 carefully curated tasks designed to extend the scope of WebArena beyond general browsing to more labor-intensive and tedious tasks. The benchmark explicitly addresses the limitations of WebArena by constructing tasks that are more challenging and suited to the current capability level of web agents. Given the rapid pace of progress and the increasing risk of benchmark saturation (Kang, 2025), we consider our effort to revisit existing benchmarks and develop new, effective ones to be a significant contribution from the benchmark construction perspective.

Our work leaves future work. (i) **Method Development**: This work primarily contributes through the construction of the benchmark and does not focus on developing new methods. We consider that designing novel methods is a crucial next step based on the findings revealed in this study. (ii) **Scalable Evaluation in Real Websites**: Although our benchmark relies on simulated environments, these environments offer a highly realistic approximation of real-world websites and were intentionally selected to guarantee reproducibility. As a next step, developing scalable evaluation methodologies for WebChoreArena in real-world environments represents a key challenge for the community.

## REPRODUCIBILITY STATEMENT

Details of the agent parameters are presented in Appendix E. The full source code is available as supplementary material. Instructions for running the code are included in the README.md.

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

APPENDIX

In this Appendix, we describe the usage of LLM for the paper in Appendix A, provide reviews of We-bArena in Appendix B, details of WebChoreArena in Appendix C, additional experiments Appendix D and experimental details in Appendix E.

## A  LLM USAGE

We employed LLMs solely as assistants for polishing the writing of this paper, and all outputs were subsequently reviewed and revised by us. Moreover, although LLMs are utilized to run agents for evaluation purposes, the core of our research lies in the manual construction of the benchmark.

## B  REVIEW OF WEBARENA

### B.1  ANNOTATION ERROR ANALYSIS

To investigate the upper bound of WebArena performance, we analyzed the annotation errors (including ambiguous task descriptions) in the WebArena benchmark. Here, regarding the errors in WebArena, some of them have already been partially documented in the AgentOccam paper (Yang et al., 2025). We therefore do not regard our review of these errors as a contribution, but rather as a necessary step to correctly estimate the upper bound in our own study.

We first extracted 229 tasks out of 684 (excluding map website) that were failed by all three Browser-Gym agents (GPT-4o, Claude 3.7 Sonnet, and Gemini 2.5 Pro). These tasks were reviewed by the authors. Our analysis revealed that approximately 134 out of 229 tasks (58.5%, 20.0% for all tasks) contained either annotation errors (75) or evaluation issues (59). As for common evaluation issues, one example is using exact_match (perfect matching with GT) without clearly instructing the agent to return only the answer string, leading to mismatches due to extra context in the output. Another example is using fuzzy_match (*i.e.,* GPT-based evaluation) without explicit formatting instructions, leading to cases where even GPT marks the agent's output incorrect due to superficial differences in format. Annotation errors were identified across all websites: 15 in Shopping, 21 in Shopping Admin, 19 in GitLab, 12 in Reddit, and 8 in Cross-site tasks.

As a result, performance would get stuck around 80%, and these issues can introduce noise that prevents the benchmark from accurately capturing the agent's true performance. This highlights the need for new benchmarks that are more challenging and carefully designed to minimize errors for the recent advanced LLM-based agents.

### B.2  RATIONALE FOR EXCLUDING THE MAP DOMAIN

As noted in Section 3.4 of the main paper, the map website in WebArena has had issues since April 2025 (*e.g.,* search results for locations no longer appear). There is an issue with this error on GitHub, and several followers raised similar concerns. Consequently, we decided not to include tasks on the map category in our WebChoreArena. However, we emphasize that inaccessibility is not the sole reason for its removal. We analyzed map websites/tasks when access was still available, and identified several critical issues that motivated the removal of the category. The main concerns are as follows:

**Limited Interaction Diversity.** The core functionality of the map website is fundamentally restricted to two actions: (i) searching for a location and (ii) finding a route between two locations. In particular, for case (i), the task of finding "B near A" is especially problematic, as the system only works when queries follow a specific format, such as specifying the amenity type followed by "near" and a location. For example, "cafe near NYU" (where cafe is a valid amenity type) works correctly, but queries like "Starbucks near NYU" (where Starbucks is not an amenity type), or "cafe close to NYU" (which does not follow the required format), do not. This significantly limits the diversity of tasks that can be constructed and makes it difficult to create challenging problems that recent LLMs-based agents struggle with. Also, many of the existing WebArena tasks in the map domain rely heavily on knowledge-based queries. For example, in the task "*Tell me the full address of all international airports that are within a driving distance of 30 km to Carnegie Art Museum*", the key challenge

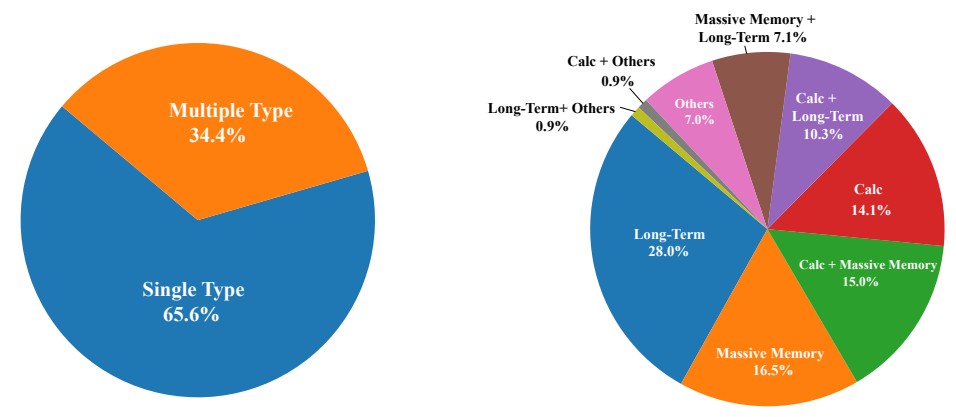

(a) Distribution of single type and multiple types     (b) Distribution of task type combinations

Figure A: Task type distributions in WebChoreArena.

| Category | Solvable Tasks | Required Input |
|---|---|---|
| Solvable with any observation | 451 | A11y Tree |
| Solvable with A11y Tree only | 69 | A11y Tree |
| Solvable with Screenshot | 12 | Screenshot + A11y Tree |

Table A: Required input modality.

should lie in retrieving locations within a 30 km radius through browsing. However, due to the above technical difficulty, agents resorts to using its own parametric knowledge to identify relevant airports and then answers the question correctly by generating their full names. We argue that such behavior does not reflect true browsing capabilities. Therefore, constructing high-quality tasks that genuinely test browsing ability in the map domain remains difficult.

**Low Reproducibility.** The map interface exhibits high sensitivity to minor input differences. First, we observed that using abbreviations versus full names for the same location often yields different results. For example, searching for "CMU" and "Carnegie Mellon University" returns slightly different locations. Also, we observed that the search outcomes can be affected by the visible region of the map at the search time.

Due to the above reasons, we decided not to create tasks in the map website. We consider that even without the map domain, the remaining four main websites are sufficient to accurately evaluate agent performance.

## C   DETAILS AND FAILURE CASES OF WEBCHOREARENA

### C.1   DETAILS OF TASK DISTRIBUTION

As described in Section 3.3, each task in WebChoreArena is associated with up to two task types. The distribution of these types is shown in Figure A. Here, since there was no significant difference between 'type_main' and 'type_sub', we counted them equally without distinguishing between primary and secondary types. Figure A (a) shows that 66.5% of the tasks belong to a single type, while the remaining tasks are associated with multiple types. Furthermore, Figure A (b) illustrates the diverse combinations of task types. These results highlight the variety and richness of tasks in WebChoreArena.

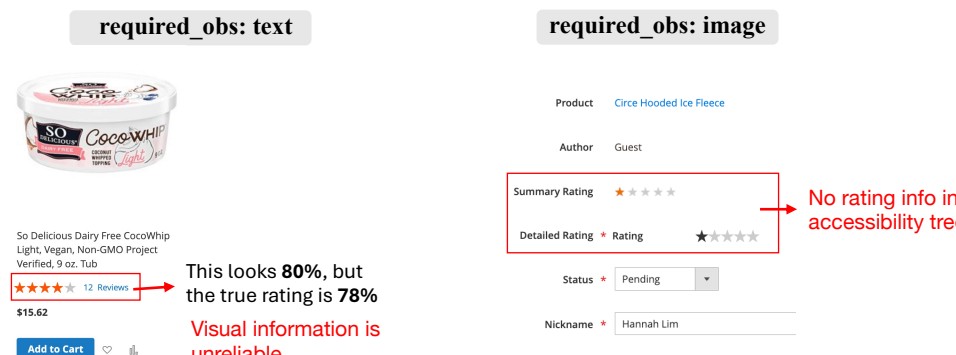

Figure B: Examples of tasks with discrepancies between visual and textual information.

## C.2  EXAMPLES OF TASKS THAT ONLY BE SOLVABLE WITH TEXT OR REQUIRE VISUAL INPUT

Figure B presents illustrative examples of tasks that only be solvable with text (accessibility tree) or require visual input (screenshot) in addition to text. As reported in Section 3.3, WebChoreArena contains 451 tasks solvable with any observation, 69 tasks that can only be solved using text input, and 12 tasks that require images. The details in Figure B are as follows:

- Left (Shopping): A product review snippet shows a rating that appears to be "80 (4 stars)" in the screenshot, but the accessibility tree reveals that the true value is "78". Due to this discrepancy, the task requires the textual accessibility tree. We label it as `required_obs` = text.
- Right (Shopping Admin): The number of review stars is shown only as graphical icons and does not appear in the accessibility tree. In this case, visual information is necessary to answer correctly, so we set `required_obs` = image.

Table A shows the required input modality for each task. For each task, we provide only the accessibility tree as input when the `required_obs` is any or text, and both the accessibility tree and a screenshot when the `required_obs` is image, as described in Section 5.2.

## C.3  FAILURE EXAMPLES IN WEBCHOREARENA

We present two failure cases per website, taken from the results of GPT-5 using BrowserGym.

### C.3.1  SHOPPING

**Failed Task1.** "*Tell me the highest-rated product in Home Audio of Electronics within the price range of $1,000.00 to $4,999.99. If the average rating is the same, select the product with the highest number of reviews. Output only the full name of the product. If there is more than one matching product, separate them with commas in ascending order of price (e.g., product1, product2).*"

**Reason for Failure1.** The agent outputs an answer without exploring all the necessary pages.

**Failed Task2.** "*Among the products made by leather not sandals in the 'Men's Shoes' category, please tell me the cheapest one with at least 5 reviews and an average rating of 4 or higher out of 5 stars. Answer with SKU number.*"

**Reason for Failure2.** The agent consumes steps by repeatedly exploring the same products.

### C.3.2  SHOPPING ADMIN

**Failed Task1.** "*What is the difference in average price between the top 10 most recently updated Simple Products with the Attribute Bag and those with the Attribute Gear? Here, if the same updated time appears beyond the top 10, include those as well. If a product with the same ID appears more*

*than once, it is a bug, so please ignore it. Do all calculations in decimals and round the purchase amount to the nearest integer just before answering.*"

**Reason for Failure1.** The agent fails to sort the data in recent update order.

**Failed Task2.** "*What is the average rating (including pending and not approved review) for all bra products? The product name must contain "bra". The rating is a value between 0 and 100 and final answer should be rounded to the nearest integer. Final answer should be "average_rating" without reasoning.*"

**Reason for Failure2.** The agent makes a tab selection error, filtering under Marketing > User Content > All Reviews instead of efficiently viewing under Reports > Reviews > By Products.

### C.3.3    REDDIT

**Failed Task1.** "*I want to analyze the top 600 most recent submissions from my account. Please count how many times I've posted in each forum, and return the result in the following dictionary format, sorted by the number of posts in descending order. If multiple forums have the same number of posts, sort them alphabetically by forum name: forum3: 51, forum1: 25, forum2: 4, forum6: 4,....*"

**Reason for Failure1.** The agent is unable to correctly count the top 600, so cannot reach the correct answer.

**Failed Task2.** " *What is the total number of upvotes that Gonzostewie's comments have received?* "

**Reason for Failure2.** The agent takes notes in Memory but uses them incorrectly.

### C.3.4    GITLAB

**Failed Task1.** " *Identify the most active contributor in facebook/create-react-app during 2021 and 2022 and show the number of commits during that period. Show the results in the following format: (contributor name), (number of commits) commits* "

**Reason for Failure1.** The agent ignores the instruction to focus on "during 2021 and 2022" when searching for answers.

**Failed Task2.** " *Tell the project that meets the following condition: the one with the most open issues among all the projects on the website. Only return the (author)/(project_name) part.* "

**Reason for Failure2.** The agent produces an answer after checking only up to the middle of the pages.

## D    ADDITIONAL EXPERIMENTAL RESULTS

### D.1    CHECKPOINT EVALUATION

To enable more fine-grained evaluation, we provide a checkpoint-based metric. This metric indicates whether the agent has correctly navigated to the required page. For example, in a management website such as Shopping Admin, it reflects whether the agent successfully reaches the product-list page when the task requires operating on that page.

For each task across the four webpages, we define a corresponding `checkpoint_url`. Table B shows whether the agent successfully visited these URLs.

From these results, we observe that models with lower overall performance on WebChoreArena, such as GPT-4o, are still able to reach the necessary pages for solving the tasks to some extent. However, their ability to carry out the core chore tasks is extremely weak, which ultimately leads to poor performance. We also find that Gemini 2.5-Pro is better than GPT-5 at reaching the correct webpages, yet it performs worse than GPT-5 on the crucial chore-tasks.

Table B: **Checkpoint evaluation.** We report the results on WebChoreArena (Full) and Checkpoint.

| Model | Shopping | | Admin | | Reddit | | GitLab | |
|---|---|---|---|---|---|---|---|---|
| | Full | Checkpoint | Full | Checkpoint | Full | Checkpoint | Full | Checkpoint |
| GPT-4o | 4.3 | 81.1 | 2.3 | 65.2 | 5.5 | 65.8 | 3.9 | 55.7 |
| Gemini2.5Pro | 31.6 | 84.2 | 43.9 | 73.5 | 34.1 | 89.5 | 37.0 | 77.1 |
| GPT-5 | 43.6 | 83.2 | 61.4 | 73.5 | 44.0 | 82.9 | 48.8 | 73.0 |

Table C: Performance with additional models and agents.

| Model | Shopping | Admin | Reddit | GitLab |
|---|---|---|---|---|
| OpenHands Cloud | 28.0 | – | 35.0 | 46.4 |
| Claude Opus 4.1 w/ Reasoning (BrowserGym) | 40.0 | 41.4 | 25.0 | 38.4 |
| DeepSeek (BrowserGym) | 36.0 | 27.6 | 25.0 | 39.3 |

## D.2 EVALUATION ON ADDITIONAL AGENTS AND MODELS

We additionally include two strong newer models, DeepSeek V3.2 Reasoner (DeepSeek-AI, 2025) and Claude Opus 4.1 (Anthropic, 2025b), and, as a new agent, OpenHands (Wang et al., 2025a) (OpenHands Cloud), which is an agent orchestration framework with a particular focus on compression and condensation of observations. As for OpenHands Cloud, we follow the configuration recommended by OpenHands (Claude Sonnet 4), whose setup reflects the performance of the most widely used OpenHands deployment. We ran experiments on the 102 subset tasks used in Table 2.

Here, for the Shopping Admin (Magento Admin Panel), OpenHands Cloud encountered webpage interaction errors on certain pages that do not occur for humans or other agents, so we omitted the results. These models and agents still do not achieve satisfactory performance on our benchmark. We plan to regularly add these models and any new models to the leaderboard and continue supporting the community through maintenance and updates.

## D.3 EVALUATION ON OPEN-SOURCE MODELS

We limited our evaluation to powerful proprietary models because even GPT-4o achieves only very low performance on WebChoreArena (around 4-7%), and open-source models perform far worse, resulting in outcomes that are not informative. As an example, we present below the results for two representative open-source LMMs: Qwen2.5-7B and Qwen2.5-14B. Table D shows the results on these models. These models cannot achieve meaningful performance on WebChoreArena because they are not trained for agent use and often have limitations in context length.

## D.4 ANALYSIS ON REASONING EFFECTS

In this section, we describe how reasoning capabilities affect performance on the WebChoreArena benchmark. To investigate this, we additionally evaluate a variant of Claude Sonnet 4 with the reasoning mode off and GPT-5 model configured with low reasoning efforts. We ran experiments on the 102 subset tasks used in Table 2. Table E shows the results on these models. We observe that reasoning clearly brings substantial improvements in performance.

## D.5 EXPERIMENT IN REAL WEBSITES

We evaluated the performance gap in real-world settings. We constructed a total of 70 real-world chore tasks and measured agent performance on them. Table F shows the results in the real websites.

As a result, we found performance trends that closely mirrored those in WebChoreArena (GPT-4o < Gemini 2.5-Pro < GPT-5). However, we also found that real-world websites change their content frequently, which makes it hard to verify results, and that many websites block agent actions for security reasons. Because of these issues, it is difficult to measure progress in a stable and sustainable

Table D: Performance with Open-source LLMs.

| Model | Shopping | Admin | Reddit | GitLab |
|-------|----------|-------|--------|--------|
| Qwen2.5-7B | 0 | 0 | 0 | 0 |
| Qwen2.5-14B | 0 | 0.0 | 0.0 | 0.0 |

Table E: Performance analysis on the reasoning effort

| Model | Calculation | Massive Memory | Long-term | Others |
|-------|-------------|----------------|-----------|--------|
| Claude Sonnet 4 (without Reasoning) | 23.9 | 16.7 | 30.6 | 27.3 |
| Claude Sonnet 4 (with Reasoning) | 37.0 | 27.8 | 36.7 | 45.5 |
| GPT-5 (reasoning = low) | 17.4 | 5.6 | 22.5 | 27.2 |
| GPT-5 (reasoning = default) | 52.2 | 44.4 | 38.8 | 81.8 |

way. Therefore, we argue that a benchmark like WebChoreArena is highly valuable, as it allows accurate evaluation within simulation environments.

### D.6 EXPERIMENT WITH NOTEBOOKS

As a preliminary experiment, we conducted experiments involving the notebook functionality. Specifically, we utilized the scratchpad page provided in the WebArena environment, which enables the agent to record intermediate information during task execution. We explicitly informed the agent that *"If you have a lot of notes to keep track of, you can use the memo at <URL for Scratchpad>."*. We extracted only the tasks requiring massive memory from the four websites, excluding the cross-site setting, and evaluated performance with and without the scratchpad.

Table G shows the results on these models. We observe that the results showed no significant impact. When examining the results in detail, we found that out of these 169 tasks, GPT-5 used the scratchpad in only 12 cases and Gemini in only 29 cases. A closer inspection showed that the scratchpad was not used effectively: the instruction sometimes caused the agents to check the scratchpad unnecessarily, or after checking it, to navigate to unrelated pages and ultimately fail the task. Because of these issues, the scratchpad did not provide meaningful benefits, and therefore, we did not include these results in the main experiments.

### D.7 EXPERIMENT WITH CALCULATORS

We investigate whether the use of external tools, particularly calculators, enhances agent performance. For this experiment, we utilized a web-based calculator developed by the WebArena team, which provides a GUI-based interface that allows agents to perform arithmetic operations seamlessly. We explicitly give agents the following instruction: *"If you need to do some calculations, you can use the calculator at <URL for Calculator>."*

We extracted 215 calculation-specific tasks from Web-ChoreArena to evaluate the effectiveness of tool use. The results are presented in Table H. The results showed that the overall performance remained largely unchanged. The main reason is that the model rarely attempts to use tools. One possible reason is that recent models already possess strong calculation abilities even without relying on the calculator. A more interesting finding is that the frequency

Table H: Performance with Calculators.

| | Acc. | | #Tool |
|---------|--------|---------|-------|
| | **Normal** | **#w. Tool** | Usage |
| Claude4 | 33.0 | 33.0 | 118 |
| GPT-5 | 50.2 | 47.4 | 2 |

of calculator usage varies substantially across models. For instance, although Claude Sonnet 4 makes relatively frequent use of the calculator, GPT-5 almost never use it, instead attempting to solve the problems directly on its own. Looking at the actual use cases, we found that the agent often use the calculators for simple computations that do not require a calculator. This observation highlights the inherent difficulty of leveraging the calculator in a effective manner.

Table F: Performance on WebChoreArena and Real-world tasks.

| Model | WebChoreArena | Real |
|---|---|---|
| GPT-4o | 3.4 | 5.7 |
| Gemini2.5-Pro | 36.8 | 24.3 |
| GPT-5 | 48.3 | 31.4 |

Table G: Comparison of performance with and without scratchpad.

| Model | Wo Scratchpad | W Scratchpad | #w. Tool |
|---|---|---|---|
| Gemini2.5Pro | 35.5 | 38.5 | 29 |
| GPT-5 | 50.3 | 47.9 | 12 |

# E    EXPERIMENTAL DETAILS

## E.1    WEB AGENT CONFIGURATIONS

### E.1.1    OVERVIEW OF AGENTS

**BrowserGym.** BrowserGym (Chezelles et al., 2024) is a unified, extensible environment for developing and evaluating web agents across diverse benchmarks with standardized observation and action spaces. The action list is shown in Table I (a). Following the implementation of WebArena, we ran WebChoreArena within the BrowserGym framework using a set of 15 actions, which are composed of basic actions, tab operations, page operations, and messaging.

**AgentOccam.** AgentOccam (Yang et al., 2025) is specifically designed for the WebArena benchmark, incorporating refined observation and action spaces to better align with the pre-training data of LLMs. Moreover, it employs a planning strategy that supports branching, allowing the agent to generate alternative plans, and pruning, which eliminates suboptimal plans based on intermediate outcomes, thereby enabling more efficient and adaptive decision-making. The action list is shown in Table I (b). Following the original implementation, we ran WebChoreArena using a set of 8 actions which are composed of basic actions, page operations, workflow management, and planning actions.

### E.1.2    EXECUTION SETTINGS

**BrowserGym.** We utilize the implementation of the BrowserGym code in Agent Workflow Memory (Wang et al., 2025b). We gratefully acknowledge the authors for providing such easily reusable code. Following the existing implementation, we permit multiple actions per step. We set the maximum number of steps to 50 for all WebChoreArena tasks.

**AgentOccam.** We utilize the original implementation of the AgentOccam (Yang et al., 2025). Following the default settings, we permit only a single action per step. We set the maximum number of steps to 50 for all WebChoreArena tasks.

### E.1.3    DISCUSSIONS ABOUT AGENT SELECTION

We prioritized a thorough evaluation and insightful analysis of the above two open-source agents that are widely adopted as baselines by the community. Nevertheless, we also conducted several preliminary experiments with other agents. First, we examined the effect of modifying the action set of BrowserGym to a Python-coding–based variant. Second, we experimented with agents specialized in memory mechanisms, such as ReasonerAgent (Chiou et al., 2025). We also attempted to run experiments using closed-source agents such as OpenAI's Operator (OpenAI, 2025a), but were unable to do so because attempts to connect to the WebArena sandbox were blocked due to a failure to establish a secure connection. Finally, we conducted experiments with BrowserUse (browser-use team, 2024); however, controlling the output format proved challenging, and even upon manual inspection, the outputs frequently contained errors. This may suggest that Browser-use, while suitable

Table I: Action Spaces

(a) Action space of BrowserGym

| Action Type | Description |
|---|---|
| noop | Do nothing |
| scroll(*dir*) | Scroll up/down |
| press(*key_comb*) | Press a key combination |
| click(*elem*) | Click at an element |
| fill(*elem*, *text*) | Type to an element, |
| hover(*elem*) | Hover on an element |
| select_option(*elem*, *option*) | Select options |
| tab_focus(*index*) | Focus on $i$-th tab |
| new_tab | Open a new tab |
| tab_close | Close current tab |
| go_back | Visit the last URL |
| go_forward | Undo `go_back` |
| goto(*URL*) | Go to URL |
| send_msg_to_user(*message*) | Send a message to the user |
| report_infeasible(*reason*) | Send special message and terminate |

(b) Action space of AgentOccam

| Action Type | Description |
|---|---|
| click[*id*] | Click at an element |
| type[*id*] [*content*] | Type into an element |
| go_back | Visit the last URL |
| go_home(*URL*) | Go to the home page |
| note[*content*] | Take notes |
| stop[*answer*] | Stop with an answer |
| branch[*id*][*intent*] | Generate a new plan |
| prune[*id*][*reason*] | Restore to a previous plan |

Table J: **Preliminary experiments for the agent selection.**

| Agent | LLM | Shopping (#25) | Admin (#29) | Reddit (#20) | GitLab (#28) | Overall (#102) |
|---|---|---|---|---|---|---|
| BrowserGym | Gemini2.5 | 28.0 | 48.3 | 30.0 | 39.3 | 37.3 |
|  | GPT-5 | 44.0 | 58.6 | 40.0 | 53.6 | 50.0 |
| ReasonerAgent | Gemini2.5 | 4.0 | 31.0 | 0 | 3.6 | 10.8 |
| BrowserGym Code | GPT-5 | 20.0 | 17.2 | 25.0 | 17.9 | 19.6 |

for browsing-oriented tasks, is not well-suited for chore-style tasks such as those in WebChoreArena.

We conducted the experiments on the small set shown in Table 2 and present the results in Table J. These results show that these methods fail to outperform BrowserGym and AgentOccam. Therefore, we chose BrowserGym and AgentOccam as the baselines.

### E.2 LLM IMPLEMENTATION DETAILS

**GPT-4o.** We used the GPT-4o model provided by Azure, specifically the `GPT-4o-2024-05-13 version`. This version was chosen because newer versions of GPT-4o tend to make agents

respond prematurely Chiou et al. (2025), and the authors of that study also recommend using `GPT-4o-2024-05-13` for more stable agent behavior. For the hyperparameters, we followed the existing implementation when available. For BrowserGym, we set the temperature to 0.1 and the max new tokens to 2,000. For AgentOccam, we set the temperature to 0.5, the top-p value to 0.95, and the max tokens to 128,000.

**Claude 3.7 Sonnet.** We used `claude-3-7-sonnet-20250219`. For the hyperparameters, we followed the existing implementation when available. For BrowserGym, we set the temperature to 0.1, top-p to 0.95, and max new tokens to 2,000. For AgentOccam, we set the temperature to 0.95, the top-p to 0.95, and the max tokens to 4,096.

**Gemini 2.5 Pro.** We used `gemini-2.5-pro`. Gemini 2.5 Pro is the most advanced reasoning Gemini model, capable of solving complex problems. For the hyperparameters, we followed the existing implementation if they exist. For BrowserGym, we set the temperature to 0.1, top-p to 0.95, and the maximum number of new tokens to 8,000. For AgentOccam, we set the default Gemini values (the specific settings are unofficial).

**Claude Sonnet 4.** We used `claude-sonnet-4-20250514`. We enable the thinking parameter. We followed the default temperature (1.0). For max completion tokens, we set 8,000 for BrowserGym and the default Claude 4 value for AgentOccam.

**GPT-5.** We used the GPT-5 model provided by Azure, specifically the `GPT-5-2025-08-07 version`. We set the reasoning level to medium, considering computational cost, time, and performance. We followed the default temperature (1.0). For max completion tokens, we set 8,000 for BrowserGym and the default GPT-5 value for AgentOccam.

### E.3 PROMPT DESIGN

#### E.3.1 WEBSITE-SPECIFIC TIPS

We refer to the WebArena implementation provided by OpenAI's CUA (OpenAI, 2025a) and incorporate website-specific tips in our experiments. The tips we provide are identical in content to those used in (OpenAI, 2025a). Below, we present the actual tips applied to each website.

---

**Tips for Shopping:**

Here are tips for using this website:

- 1. This website provides very detailed category of products. You can hover categories on the top menu to see subcategories.

- 2. If you need to find information about your previous purchases, you can go My Account > My Orders, and find order by date, order number, or any other available information

- 3. An order is considered out of delivery if it is marked as "processing" in the order status

- 4. When the task asks you to draft and email. DO NOT send the email. Just draft it and provide the content in the last message

---

**Tips for Shopping Admin:**

Here are tips for using this website:

- 1. When you add a new product in the CATALOG > Products tab, you can click the downwardarrow beside the "Add Product" button to select options like "Simple Product", "Configurable Product", etc.

- 2. If you need to add new attribute values (e.g. size, color, etc) to a product, you can find the product at CATALOG > Products, search for the product, edit product with "Configurable Product" type, and use "Edit Configurations" to add the product with new attribute values. If the value that you want does not exist, you may need to add new values to the attribute.

---

- 3. If you need to add new values to product attributes (e.g. size, color, etc), you can visit STORES > Attributes > Product, find the attribute and click, and add value after clicking "Add Swatch" button.

- 4. You can generate various reports by using menus in the REPORTS tab. Select REPORTS > "report type", select options, and click "Show Report" to view report.

- 5. In this website, there is a UI that looks like a dropdown, but is just a 1-of-n selection menu. For example in REPORTS > Orders, if you select "Specified" Order Status, you will choose one from many options (e.g. Canceled, Closed, ...), but it's not dropdown, so your click will just highlight your selection (1-of-n select UI will not disappear).

- 6. Configurable products have some options that you can mark as "on" of "off". For example, the options may include "new", "sale", "eco collection", etc.

- 7. You can find all reviews and their counts in the store in MARKETING > User Content > All Reviews. If you see all reviews grouped by product, go REPORTS > By Products and search by Product name.

- 8. This website has been operating since 2022. So if you have to find a report for the entire history, you can select the date from Jan 1, 2022, to Today.

- 9. Do not export or download files, or try to open files. It will not work.

### Tips for Reddit:

Here are tips for using this website:

- 1. when the task mentions subreddit, it is referring to 'forum'.

- 2. if you need find a relevant subreddit or forum, you can find the name after clicking "alphabetical" in the "Forum" tab.

- 3. if you have to find submissions (posts) or comments by a particular user, visit reddit.site/user/<user name> to see the list

### Tips for GitLab:

Here are tips for using this website:

- 1. your user name is byteblaze

- 2. To add new members to the project, you can visit project information > members tab and click blue "invite members" button on top right

- 3. To set your status, click profile button on top right corner of the page (it's next to the question mark button) and click edit status

- 4. To edit your profile, click profile button on top right corner of the page (it's next to the question mark button) and click edit profile

- 5. You can also access to your information e.g. access token, notifications, ssh keys and more from "edit profile" page

- 6. Projects that you have contributed to are listed under Project / Yours / All tab of gitlab.site. You can sort repos using dropdown button on top right

- 7. Projects's repository tab has menus like Commits, Branches, Contributors, and more. Contributors tab shows contributors and their number of commits

- 8. If you want to see all the issues for you, you can either click button on the right of + icon on top right menu bar

- 9. When the task mentions branch main, it often means master

### E.3.2 FULL PROMPT EXAMPLES

We present examples of the full input provided to the agent in Figure C, D, E, and F. Figure C and D illustrates input examples from BrowserGym, while Figure E and F show input examples from AgentOccam.

When the task is a cross-site task, we add some hints following the original implementations. For BrowserGym, we add the following hint after the task description: "*(Note: if you want to visit other websites, check out the homepage at <home_url>. It has a list of websites you can visit. <home_url>/password.html lists all the account name and password for the websites. You can use them to log in to the websites.)*". For AgentOccam, we add the additional action after the last action description: "*- go_home: To return to the homepage where you can find other websites.*"

Based on these inputs, we conduct a discussion on the memory mechanism in the following section.

### E.4 AGENT MEMORY MECHANISMS

We explain the details of the memory function for AgentOccam and BrowserGym. BrowserGym adopts an explicit memory mechanism. At each step, the agent outputs a reason for action, the action itself, and, when necessary, memory content to be stored. In the subsequent step, the input includes the past reasons for action, actions, and memory entries. This setup allows the agent to retain and refer back to essential information in memory. Therefore, for tasks that require past memory, it is sufficient for the agent to output the relevant memory information appropriately at each step.

In contrast, AgentOccam employs a different strategy. At each step, it outputs the interaction history summary, observation summary, reason for action, action, and an observation highlight. The next input includes the reason for action, action, and the observation highlight (or the observation summary if the highlight is too long). A key feature of AgentOccam is the `note [content]` action, which allows the agent to explicitly store important information. Once this action is issued, the content within `[content]` is included in subsequent inputs.

The key difference lies in memory handling: BrowserGym explicitly outputs memory at each step, while AgentOccam relies on summaries or must issue a `note [content]` action to retain important information. This explains why AgentOccam performs worse on Massive Memory tasks in Figure 4.

You are an agent trying to solve a web task based on the content of the page and a user instructions. You can interact with the page and explore. Each time you submit an action it will be sent to the browser and you will receive a new page.

Here are tips for using this website:
1. This website provides very detailed category of products. You can hover categories on the top menu to see subcategories.
2. If you need to find information about your previous purchases, you can go My Account > My Orders, and find order by date, order number, or any other available information
3. An order is considered out of delivery if it is marked as "processing" in the order status
4. When the task asks you to draft and email. DO NOT send the email. Just draft it and provide the content in the last message

Figure C: A prompt example of `system_message` for BrowserGym.

You are a UI Assistant, your goal is to help the user perform tasks using a web browser. You can communicate with the user via a chat, in which the user gives you instructions and in which you can send back messages. You have access to a web browser that both you and the user can see, and with which only you can interact via specific commands.

Review the instructions from the user, the current state of the page and all other information to find the best possible next action to accomplish your goal. Your answer will be interpreted and executed by a program, make sure to follow the formatting instructions.

## Chat messages:
- [assistant] Hi! I am your UI assistant, I can perform web tasks for you. What can I help you with?
- [user] Please provide the distribution of reviews for Snakebyte Twin Charge X - Xbox One Controller Charger Dual Docking/Charging Station incl. 2 Rechargeable Battery Packs for XBOX One Controller / Elite / S Controller Gamepad, Black. Here, the review stars (1-5) correspond to a Rating that is 20 times their value (1 = Rating 20, 2 = Rating 40, 3 = Rating 60, 4 = Rating 80, 5 = Rating 100).
Follow the format below, using numerical values: 5: {number}, 4: {number}, 3: {number}, 2: {number}, 1: {number}

# Observation of current step:
## AXTree: (omitted)

# History of interaction with the task:
## step 0
### Action:
hover('856')

...(omitted some steps)

## step 7
### Action:
hover('1068')
click('1068')
### Memory:
The product is related to Xbox One and should be under the "Video Games" category.

## step 8
### Action:
hover('1068')
click('1068')
### Memory:
The product "Snakebyte Twin Charge X - Xbox One Controller Charger Dual Docking/Charging Station incl. 2 Rechargeable Battery Packs for XBOX One Controller / Elite / S Controller Gamepad, Black" is related to Xbox One and should be under the "Video Games" category.

# Action space:

15 different types of actions are available. (omitted)

# Abstract Example
Here is an abstract version of the answer with description of the content of each tag. Make sure you follow this structure, but replace the content with your answer:
(omitted)

# Concrete Example
Here is a concrete example of how to format your answer. Make sure to follow the template with proper tags:
(omitted)

Figure D: A prompt example of `user_message` for BrowserGym.

You are an AI assistant performing tasks on a web browser. You will be provided with task objective, current step, web page observations, previous plans, and interaction history. You need to issue an action for this step.

Generate the response in the following format:
INTERACTION HISTORY SUMMARY:
Emphasize all important details in the INTERACTION HISTORY section.
OBSERVATION DESCRIPTION:
Describe information in the CURRENT OBSERVATION section. Emphasize elements and features that are relevant or potentially helpful for fulfilling the objective in detail.
REASON:
Provide your rationale for proposing the subsequent action commands here.
ACTION:
Select your action here.
OBSERVATION HIGHLIGHT:
List the numerical ids of elements on the current webpage based on which you would issue your action. Also include elements on the current webpage you would attend to if you fail in the future and have to restore to this step. Don't include elements from the previous pages. Select elements at a higher hierarchical level if most their children nodes are considered crucial. Sort by relevance and potential values from high to low, and separate the ids with commas. E.g., '1321, 52, 756, 838'.

You are ONLY allowed to use the following action commands. Strictly adheres to the given format. Only issue one single action. If you think you should refine the plan, use the following actions:
- branch [parent_plan_id] [new_subplan_intent]: To create a new subplan based on PREVIOUS PLANS. Ensure the new subplan is connected to the appropriate parent plan by using its ID. E.g., 'branch [12] [Navigate to the "Issue" page to check all the issues.]'
- prune [resume_plan_id] [reason]: To return to a previous plan state when the current plan is deemed impractical. Enter the ID of the plan state you want to resume. E.g., 'prune [5] [The current page lacks items "black speaker," prompting a return to the initial page to restart the item search.]' Otherwise, use the following actions:
- click [id]: To click on an element with its numerical ID on the webpage. E.g., 'click [7]' If clicking on a specific element doesn't trigger the transition to your desired web state, this is due to the element's lack of interactivity or GUI visibility. In such cases, move on to interact with OTHER similar or relevant elements INSTEAD.
- type [id] [content] [press_enter_after=0|1]: To type content into a field with a specific ID. By default, the "Enter" key is pressed after typing unless 'press_enter_after' is set to 0. E.g., 'type [15] [Carnegie Mellon University] [1]' If you can't find what you're looking for on your first attempt, consider refining your search keywords by breaking them down or trying related terms.
- stop [answer]: To stop interaction and return response. Present your answer within the brackets. If the task doesn't require a textual answer or appears insurmountable, indicate "N/A" and additional reasons and all relevant information you gather as the answer. E.g., 'stop [5h 47min]'
- note [content]: To take note of all important info w.r.t. completing the task to enable reviewing it later. E.g., 'note [Spent $10 on 4/1/2024]'
- go_back: To return to the previously viewed page.

Here are tips for using this website:
1. This website provides very detailed category of products. You can hover categories on the top menu to see subcategories.
2. If you need to find information about your previous purchases, you can go My Account > My Orders, and find order by date, order number, or any other available information
3. An order is considered out of delivery if it is marked as "processing" in the order status
4. When the task asks you to draft and email. DO NOT send the email. Just draft it and provide the content in the last message

Figure E: A prompt example of `system_message` for AgentOccam.

OBJECTIVE: Please provide the distribution of reviews for Snakebyte Twin Charge X - Xbox One Controller Charger Dual Docking/Charging Station incl. 2 Rechargeable Battery Packs for XBOX One Controller / Elite / S Controller Gamepad, Black. Here, the review stars (1 5) correspond to a Rating that is 20 times their value (1 = Rating 20, 2 = Rating 40, 3 = Rating 60, 4 = Rating 80, 5 = Rating 100). Follow the format below, using numerical values: 5: {number}, 4: {number}, 3: {number}, 2: {number}, 1: {number}

PREVIOUS PLANS: [0] (Active Plan) Find the solution to "Please provide the distribution of reviews for Snakebyte Twin Charge X - Xbox One Controller Charger Dual Docking/Charging Station incl. 2 Rechargeable Battery Packs for XBOX One Controller / Elite / S Controller Gamepad, Black. Here, the review stars (1 5) correspond to a Rating that is 20 times their value (1 = Rating 20, 2 = Rating 40, 3 = Rating 60, 4 = Rating 80, 5 = Rating 100). Follow the format below, using numerical values: 5: {number}, 4: {number}, 3: {number}, 2: {number}, 1: {number}"

INTERACTION HISTORY:
<step_0_interaction>
OBSERVATION:
RootWebArea 'One Stop Market'
  link 'My Account'
  link 'My Wish List'
  link 'Sign Out'
  text 'Welcome, Emma Lopez!'
  link 'Skip to Content'
  link 'store logo'
  link 'My Cart'
  combobox 'Search' [required: False]
  link 'Advanced Search'
  button 'Search' [disabled: True]
REASON FOR ACTION:
To locate the specific product and its review distribution, I need to use the search functionality provided on the webpage. This will help narrow down the search to the exact product.
ACTION:
type [2139] [Snakebyte Twin Charge X - Xbox One Controller Charger Dual Docking/Charging Station incl. 2 Rechargeable Battery Packs for XBOX One Controller / Elite / S Controller Gamepad, Black] [1]
</step_0_interaction>
<step_1_interaction>
OBSERVATION:
The search results page for "Snakebyte Twin Charge X - Xbox One Controller Charger Dual Docking/Charging Station incl. 2 Rechargeable Battery Packs for XBOX" is displayed. The relevant product is listed with the following details: - Product link: 'Snakebyte Twin Charge X - Xbox One Controller Charger Dual Docking/Charging Station incl. 2 Rechargeable Battery Packs for XBOX One Controller / Elite / S Controller Gamepad, Black' - Rating: 52 - 32 Reviews There are also options to add the product to the cart, wish list, or compare it with other products.
REASON FOR ACTION:
To view the distribution of reviews for the specific product, I need to click on the product link to access its detailed review section.
ACTION:
click [6276]
</step_1_interaction>
CURRENT OBSERVATION:
(omitted)

Figure F: A prompt example of `user_message` for AgentOccam.

