# OpenReview forum: "WebChoreArena: Evaluating Web Browsing Agents on Realistic Tedious Web Tasks"
_ICLR.cc/2026/Conference — Submitted to ICLR 2026_

### Official Review · Reviewer_HpZT · 2025-10-25

**Soundness:** 3
**Presentation:** 3
**Contribution:** 3
**Rating:** 4
**Confidence:** 4

**Summary:**

This paper introduces **WebChoreArena**, a new benchmark designed to systematically evaluate LLM-powered web agents on tedious, memory-intensive, and reasoning-demanding web tasks. The benchmark consists of 532 manually constructed tasks across four simulated WebArena environments — *Shopping*, *Shopping Admin*, *Reddit*, and *GitLab* — totaling over 300 hours of human annotation. Tasks are categorized into Massive Memory, Calculation, Long-Term Memory, and Others, enabling comprehensive assessment of agents’ cognitive and operational capabilities. Fully compatible with WebArena for reproducibility and comparability, the authors evaluate AgentOccam and BrowserGym frameworks using several state-of-the-art LLMs, including GPT-4o, Claude Sonnet 3.7/4, Gemini 2.5 Pro, and GPT-5.

**Strengths:**

The paper presents a well-motivated and timely benchmark that addresses an important gap in evaluating LLM-based web agents on complex, tedious, and memory-intensive tasks. It demonstrates strong methodological rigor, with 532 carefully curated tasks covering diverse domains and requiring multi-step reasoning, long-term memory, and precise calculation. The benchmark’s full compatibility with WebArena ensures reproducibility and enables fair cross-agent comparison. The authors conduct comprehensive experiments using multiple state-of-the-art frameworks (AgentOccam and BrowserGym) and several leading LLMs (GPT-4o, GPT-5, Claude, Gemini), providing valuable insights into model weaknesses and task-specific performance trends. In addition, the clarity of writing, systematic task taxonomy, and inclusion of human baselines make the paper highly accessible and informative. Overall, WebChoreArena is a significant and practical contribution that is likely to become an essential resource for future research on web agents and long-horizon reasoning.

**Weaknesses:**

First, the study lacks algorithmic innovation, focusing solely on benchmark construction rather than proposing new agent mechanisms. Second, all experiments are confined to simulated environments, which limits the ecological validity and generalizability to real-world dynamic websites. Third, the model coverage is insufficient — only a few large proprietary models (e.g., GPT and Claude series) are tested, with no inclusion of diverse model sizes or more open-source models (such as LLaMA, Qwen, or Mistral families). Expanding the evaluation to a broader spectrum of model scales and architectures would make the benchmark more representative and strengthen its conclusions. Finally, the error analysis remains descriptive without suggesting concrete improvements, and the benchmark’s high manual construction cost may hinder scalability.

**Questions:**

1. Model Diversity:
The paper evaluates several leading models (GPT-4o, GPT-5, Claude, Gemini), but the coverage remains limited. Could the authors expand the experiments to include a wider variety of both open-source and proprietary models, such as LLaMA, Qwen, Mistral, or DeepSeek, to better understand performance trends across different architectures and scales?

2. Model Scaling Analysis:
Have the authors examined how model size or reasoning depth affects performance on different task types (e.g., Massive Memory vs. Calculation)? Including a scaling curve analysis could provide valuable insights into whether task success correlates with model capacity or training methodology.

3. Closed-Source Model Comparison:
Beyond GPT and Claude, are there plans to evaluate other strong closed-source models, such as Gemini Ultra, Claude Opus, or proprietary enterprise agents? This would help position WebChoreArena as a more comprehensive benchmark for the broader LLM ecosystem.

4. Open-Source Baseline Integration:
Would the authors consider including smaller open-source baselines (e.g., 7B–14B models) to establish a clearer performance hierarchy and make the benchmark more accessible for the academic community?

5. Generalization Beyond Simulation:
Since the current setup relies on simulated WebArena environments, do the authors plan to extend the benchmark to real-world dynamic websites and test whether the same models maintain consistent performance under non-deterministic conditions?

---

> ### Author Response · Authors · 2025-11-22
> **Rebuttal1/2**
>
> We sincerely appreciate that the reviewer has thoroughly engaged with our work and highlighted several strengths, including the clear motivation of the benchmark, the rigorous and carefully curated task design, and full compatibility with WebArena for reproducibility. We are also grateful for the reviewer’s recognition of the clarity of the paper and the benchmark’s potential value to the community.
>
> We have carefully considered your suggestions and updated the manuscript accordingly, with changes highlighted in magenta. Below, we provide detailed responses to each of your points:
>
> *Note: Unless otherwise specified, the mentioned line numbers denote those of the original submitted paper version instead of the rebuttal revision version.*
>
> > ### **W1. Contribution focused on benchmark development**
>
> Thanks for the question. While we do not develop an innovative agent, we believe that the construction of a high-quality benchmark and its thorough analysis constitute a substantial contribution to the Datasets and Benchmarks area at ICLR. In recent years, the rapid evolution of models has led to benchmark saturation and growing difficulty in ensuring benchmark quality. Therefore, developing high-quality benchmarks is increasingly important, as evidenced by the large number of benchmark-focused papers published at ICLR 2025.
>
> We spent more than 300 hours ensuring the quality of the benchmark and designing it to push the boundary of the existing agents' capabilities.
> Furthermore, as described in L424–431, our well-designed task taxonomy enables fine-grained analysis of task-specific performance and helps reveal the strengths and weaknesses of different agent mechanisms.
>
> We believe that this dataset and the detailed analyses provide an important contribution to ICLR, even without proposing a novel agent method.
>
> > ### **W3, Q1, Q4: Evaluation with Open-source LLMs**
>
> Thank you for the feedback. We limited our evaluation to powerful proprietary models because even GPT-4o achieves only very low performance on WebChoreArena (around 4–7%), and open-source models perform far worse, resulting in outcomes that are not informative.
>
> As an example, we present below the results for two representative open-source LMMs: Qwen2.5-7B and Qwen2.5-14B.
>
> |  | Shopping | admin | reddit | gitlab |
> | --- | --- | --- | --- | --- |
> | Qwen2.5-7B | 0 | 0 | 0 | 0 |
> | Qwen2.5-14B | 0 | 0.0 | 0.0 | 0.0 |
>
>
> These models cannot achieve meaningful performance on WebChoreArena because they are not trained for agent use and often have limitations in context length.
>
> > ### **Q2. The analysis of reasoning depth**
>
> Thank you for the suggestion. We have examined how reasoning affects performance on WebChoreArena. Here, we have not conducted model-size analysis because open-source models fail to produce meaningful performance, and the model sizes of proprietary systems are not publicly disclosed. We ran experiments on the 102 subset tasks used in Table 2.
>
> |  | Calculation | Massive Memory | Long-term | Others |
> | --- | --- | --- | --- | --- |
> | Claude Sonnet4 wo Reasoning | 23.9 | 16.7 | 30.6 | 27.3  |
> | Claude Sonnet4 w Reasoning | 37.0 | 27.8 | 36.7 | 45.5 |
> | GPT-5 (reasoning=low) | 17.4 | 5.6 | 22.5 | 27.2 |
> | GPT-5 (reasoning=default) | 52.2 | 44.4 | 38.8 | 81.8 |
>
> We observe that reasoning clearly brings substantial improvements in performance.
>
> > ### **W3, Q1, Q3. Results with additional models and agents**
>
> Thanks for the suggestion. We additionally have included two strong newer models, DeepSeek and Claude Opus 4.1, and, as a new agent, OpenHands (OpenHands Cloud), which is an agent orchestration framework with a particular focus on compression and condensation of observations.
> Here, the model mentioned by the reviewer as “Gemini Ultra” is not currently accessible, so we did not evaluate it.
> We ran experiments on the 102 subset tasks used in Table 2.
> The results are shown below.
>
> |  | Shopping | Admin | Reddit | GitLab |
> | --- | --- | --- | --- | --- |
> | OpenHands | 28.0 | -* | 35.0 | 46.4 |
> | Claude Opus4.1 w Reasoning (BrowserGym) | 40.0 | 41.4 | 25.0 | 38.4 |
> | DeepSeek (BrowserGym) | 36.0 | 27.6 | 25.0 | 39.3 |
>
> *For the Shopping Admin (Magento Admin Panel), OpenHands Cloud encountered webpage interaction errors on certain pages that do not occur for humans or other agents, so we omitted the results.*
>
> These models and agents still do not achieve satisfactory performance on our benchmark.
> We plan to regularly add these models and any new models to the leaderboard and continue supporting the community through maintenance and updates.

---

> > ### Author Response · Authors · 2025-11-22
> > **Rebuttal2/2**
> >
> > > ### **W4. Does the error analysis fail to provide concrete suggestions for improvement?**
> >
> > Thanks for the question. Although Section 5 may appear to simply summarize failure cases without providing explicit suggestions, our main findings, particularly Finding 3, do highlight what is needed to improve performance on WebChoreArena.
> > Specifically, as described in Lines 424–431, our well-defined task taxonomy enables fine-grained, type-specific performance analysis and reveals how different agent mechanisms work under each task type. As explained in Appendix E.4, the analysis makes clear that these differences can be attributed to the memory mechanism of each agent.
> >
> > Therefore, we believe that our benchmark encourages improvements in future research efforts.
> >
> > > ### **W4.  The benchmark’s high manual construction cost may hinder scalability.**
> >
> > While it is true that our benchmark construction lacks scalability, we believe that our carefully constructed and validated benchmark is highly advantageous for assessing accurate progress.
> >
> > Many influential benchmarks, such as MMLU [1], MMMU [2],  OSWorld [3], Humanity’s Last Exam [4], were also created through substantial human effort. These works show that human-designed tasks are essential for ensuring clarity, correctness, and meaningful difficulty.
> > In contrast, relying solely on automated benchmark generation can easily introduce ambiguous instructions, incorrect ground-truth labels, or tasks that fail to meaningfully challenge stronger models. Such issues often lead to unstable or misleading evaluations.
> >
> > Although the scalability is important for training data, which must be generated at a large scale, evaluation data should instead prioritize accurate and reliable assessment.
> >
> > We believe that careful human design and validation are essential for accurately measuring progress.
> >
> > [1] Hendrycks+, Measuring Massive Multitask Language Understanding, ICLR2021
> >
> > [2] Yue+, MMMU: A Massive Multi-discipline Multimodal Understanding and Reasoning Benchmark for Expert AGI, CVPR2024
> >
> > [3] Xie+, OSWorld: Benchmarking Multimodal Agents for Open-Ended Tasks in Real Computer Environments, NeurIPS2024 Dataset and Benchmark Track
> >
> > [4] Phan+, Humanity's Last Exam, arXiv2025
> >
> > > ### **W2. Q5: Does the use of a simulation environment limit the benchmark’s generalizability to real-world, dynamic websites?**
> >
> > While real-world evaluation is certainly important, we believe that evaluation in simulation environments offers certain advantages that real-world evaluation cannot provide.
> >
> > We argue that our evaluation in the WebArena simulation environments allows accurate and reproducible evaluation, while minimizing the gap to real-world, dynamic websites.
> >
> > 1. **The simulation environment enables accurate evaluation.**
> >
> > We prioritized constructing the tasks in a simulation environment to ensure accurate and sustainable measurement of progress. Within this setting, we instructed annotators to create queries that reflect plausible real-world use cases, rather than simply making the tasks artificially difficult.
> >
> > Among simulation environments, the WebArena environment faithfully reproduces real-world webpages such as shopping sites, admin panels, GitLab, and Reddit, and therefore inherently exhibits a relatively small sim-to-real gap. As a result, the findings in this study should generalize well to real-world settings.
> >
> > 2. **The gap between simulation and real-world performance is small**
> >
> > We agree that it is also important to carefully verify the gap between the real world and the simulation environment. To address this, we evaluated the performance gap in real-world settings.
> >
> > We constructed a total of 70 real-world chore tasks and measured agent performance on them.
> >
> > |  | WebChoreArena | Real |
> > | --- | --- | --- |
> > | GPT-4o | 3.4 | 5.7 |
> > | Gemini2.5-Pro | 36.8 | 24.3 |
> > | GPT-5 | 48.3 | 31.4 |
> >
> > As a result, we found performance trends that closely mirrored those in WebChoreArena (GPT-4o < Gemini 2.5-Pro < GPT-5).
> >
> > However, we also found that real-world websites change their content frequently, which makes it hard to verify results, and that many websites block agent actions for security reasons. Because of these issues, it is difficult to measure progress in a stable and sustainable way.
> >
> > Therefore, we argue that a benchmark like WebChoreArena is highly valuable, as it allows accurate evaluation within environments.

---

### Official Review · Reviewer_5TKV · 2025-10-26

**Soundness:** 3
**Presentation:** 3
**Contribution:** 3
**Rating:** 4
**Confidence:** 4

**Summary:**

Given the complexity limitations of current agent benchmarks, this work proposes a new WebChoreArena benchmark, that features more challenging tasks stressing massive memory, calculation, and long-term memory handling. Experimental results with various agents reveal directions for future agent development.

**Strengths:**

1. **Benchmarking Contribution to the Agent Community.**
> Regarding recent agent progress. As existing mainstream benchmarks are beginning to be solved, this work introduces more complex tasks, particularly targeting massive memory, calculation, and long-term memory handling scenarios, to facilitate the development of more capable agents.

2. **High-quality example curation process.**
> The tasks are manually collected by agent researchers. Measures of task properties are also reported.

3. **Decent benchmarking effort.**
> This paper benchmarks multiple major open-source agent frameworks, with a series of LM backbones. Further, multiple analyses were conducted to offer more insights.

**Weaknesses:**

1. **Lack of Fine-Grained Evaluation.**
> As the tasks become more complex (i.e., involve more checkpoints that agents need to achieve), a single end-task evaluation tells less information. It would be more informative if more fine-grained intermediate evaluations, especially targeting each atomic requirement in the task instructions, were included in the benchmark.

2. **Unclear in complexity?**
> Although tasks are claimed to be more challenging than the WebArena benchmark, the best agent (with GPT-5) already achieves ~50% success rate, indicating that the benchmark may be largely solved by the frequently-updated models soon (?). Further on this point, it is very likely that the two agent frameworks experimented in this work do not bring out the best performance of GPT-5 (as opposed to ChatGPT or other popular agent frameworks), therefore it is possible that some existing agents already can solve 60-70% of this benchmark.

**Questions:**

1. How realistic are the tasks in WebChoreArena, and what processes have been applied to ensure the realism of tasks? As opposed to overly driving up the task complexity, potentially to a point where humans wouldn’t even need to do such tasks.
On the other hand, is the purpose of this benchmark to: (i) reflect how agents perform in practice, (ii) act as stress tests to agents, or others?

---

> ### Author Response · Authors · 2025-11-22
> **Rebuttal1/2**
>
> We sincerely appreciate that the reviewer has taken the time to thoroughly understand our work and to highlight several strengths, including the benchmarking contribution to the agent community, a high-quality example curation process, and a decent benchmarking effort.
>
> We have carefully considered your suggestions and updated the manuscript accordingly, with changes highlighted in magenta. Below, we provide detailed responses to each of your points:
>
> *Note: Unless otherwise specified, the mentioned line numbers denote those of the original submitted paper version instead of the rebuttal revision version.*
>
> > ### **W1. Lack of Fine-Grained Evaluation.**
>
> Thank you very much for this important suggestion.  We consider that our benchmark enables a more fine-grained evaluation.
>
> In the task configuration (provided in the Supplementary Materials), the `start_url_lite` field specifies the URL of a page that the agent must pass through in order to solve the task. In practice, this page serves as a `checkpoint_url` that indicates whether the agent has correctly navigated to the required location. For example, in a management website such as Shopping Admin, it shows whether the agent successfully reaches the product-list page when the task requires working on that page.
>
> Whether the agent visits the checkpoint_url accurately reflects its ability to navigate to the correct page before performing the key chore tasks, which allows us to more precisely diagnose the sources of errors for each agent.
>
> We present these results below.
>
> |  | Shopping | Admin | Reddit | GitLab |  |
> | --- | --- | --- | --- | --- | --- |
> | GPT-4o |  4.3 / 81.1 |    2.3 / 65.2 |   5.5 / 65.8 |    3.9 / 55.7 |  |
> | Gemini2.5Pro | 31.6 / 84.2  | 43.9 / 73.5 | 34.1  / 89.5 |   37.0 / 77.1 |  |
> | GPT-5 | 43.6 / 83.2 | 61.4 / 73.5 | 44.0 / 82.9 | 48.8 / 73.0 |  |
>
> From these results, we observe that models with lower overall performance on WebChoreArena, such as GPT-4o, are still able to reach the necessary pages for solving the tasks to some extent. However, their ability to carry out the core chore tasks is extremely weak, which ultimately leads to poor performance.
>
> We also find that Gemini 2.5-Pro is better than GPT-5 at reaching the correct webpages, yet it performs worse than GPT-5 on the crucial chore-tasks.
>
> In the final version, we plan to rename `start_url_lite` in the task config to `checkpoint_url`.
>
> > ### **W2.1. Are the agents used in the evaluation unable to fully extract the capability of the current models?**
>
> Thanks for the question. We would like to clarify that the agents used in our study, including BrowserGym and AgentOccam, are among the most capable open-source agents currently available. As noted in Lines 353–355, they provide powerful and reproducible baselines for the open-source research community. Therefore, we believe that these agents effectively assess the models’ capabilities with open-source agents, which is essential for the academic research community such as ICLR.
>
> As for other agents, we conducted preliminary evaluations using popular agent frameworks such as BrowserUse and verified that they exhibit frequent errors, such as counting mistakes, when handling chore-style tasks in WebChoreArena (Lines 939–940). Furthermore, although we report results for a more orchestrated agent like OpenHands in the rebuttal (in reviewer cMr3’s thread), their performance remains around 40%, indicating that they are still far from adequate.
>
> In addition, although we cannot evaluate ChatGPT on WebChoreArena due to security restrictions that block access to WebArena environments, internal developers at OpenAI have used the original WebArena to assess ChatGPT’s agentic capabilities [1]. This indicates that there is an even greater need for a more challenging benchmark, such as WebChoreArena.
>
> [1] OpenAI, Introducing ChatGPT agent: bridging research and action, https://openai.com/index/introducing-chatgpt-agent/
>
> > ### **W2.2. Given that current models already attain close to 50% accuracy, could future models solve the benchmark rapidly?**
>
> Thanks for the question. While it might be possible that future models may achieve higher scores, we believe that WebChoreArena will continue to offer meaningful headroom for evaluation.
>
> Although GPT-5 achieves roughly 50% on WebChoreArena, we consider that this does not mean saturation, and it indicates that there is substantial room for improvement. Unlike WebArena, which includes ~20% ambiguous tasks and can effectively measure performance only up to ~80%, WebChoreArena removes such ambiguities through extensive human verification.
> As a result, even when future agents achieve 60–70%, substantial headroom remains.

---

> ### Author Response · Authors · 2025-11-22
> **Rebuttal2/2**
>
> > ### **Q1. Realism of the tasks and the purpose of this benchmark**
>
> Thanks for your question. The goal of our tasks is to evaluate tasks that humans actually want to automate using agents.
>
> The demand for the chore tasks covered in WebChoreArena is increasing, as demonstrated by studies conducted on real workers showing that a growing number of them seek automation for tedious and repetitive chores [2] (L96–97).
>
> Indeed, Figure 1 in the main paper illustrates this point by depicting tasks that are familiar to humans yet still labor-intensive and demanding. The significance of these tasks has also been underscored by Reviewer cMr3 and Reviewer 2m48.
>
> [2] Shao+, Future of Work with AI Agents: Auditing Automation and Augmentation Potential across the U.S. Workforce, arXiv2025

---

### Official Review · Reviewer_2m48 · 2025-11-01

**Soundness:** 3
**Presentation:** 3
**Contribution:** 3
**Rating:** 4
**Confidence:** 4

**Summary:**

The authors introduce WebChoreArena, a benchmark of 532 human curated web agent tasks designed to be more tedious and complex than WebArena tasks. These tasks feature three challenge types: massive memory, calculation, long term memory. The paper evaluated leading models including GPT-4o, Claude 3.7 Sonnet, Gemini 2.5 Pro with two agent frameworks of AgentOccam and BrowserGym.

**Strengths:**

The paper shows clear problem focus and task taxonomy: benchmark that targets tedious chores underrepresented in prior work.

The paper builds on top of existing, proven sandbox environments of WebArena, saving efforts for adaptation and potential environment pitfalls.

Annotators followed explicit guidelines to curate tasks, with cross checking to minimize labeling, evaluation errors.

**Weaknesses:**

It would be helpful to see more analysis on agent’s use of calculation related tools in completing requirements, on top of the fact that models don’t always choose to use calculators.

Has the author examined if models are given sufficient tools or agentic design that enables long term memory. An example could be a notebook page, or a function call that allows models to write/read/search from a notebook. It would be great to see some analysis on how well models utilize these functions to complete tasks that require longer term memory.

It would be great to provide more details on how different modality is used for the agent’s action.

**Questions:**

Listed above.

---

> ### Author Response · Authors · 2025-11-22
> **Rebuttal**
>
> We sincerely appreciate that the reviewer has taken the time to thoroughly understand our work and to highlight several strengths, including the clear problem focus and task taxonomy,  the importance of building on the WebArena environments,  the practical focus on chore-like tasks, and a meticulous curated process.
>
> We have carefully considered your suggestions and updated the manuscript accordingly, with changes highlighted in magenta. Below, we provide detailed responses to each of your points:
>
> *Note: Unless otherwise specified, the mentioned line numbers denote those of the original submitted paper version instead of the rebuttal revision version.*
>
> > ### **W1. More analysis when using the calculator**
>
> Thanks for the suggestion. Looking at the actual use cases, we found that the agent often uses the calculators for simple computations that do not require a calculator. This observation highlights the inherent difficulty of leveraging the calculator in an effective manner.
>
> > ### **W2. Experiments using the Notebook page**
>
> Thanks for your question. While we agree that developing more capable agents that utilizes notebook pages is indeed important, we believe determining how to design more sophisticated note-taking mechanisms remains an open challenge for future research, building on WebChoreArena. We expect WebChoreArena to serve as a solid foundation for such subsequent work.
>
> As a preliminary experiment, we conducted experiments involving the notebook functionality. Specifically, we utilized the scratchpad page provided in the WebArena environment, which enables the agent to record intermediate information during task execution. We explicitly informed the agent that "If you have a lot of notes to keep track of, you can use the memo at {scratchpad url}.".We extracted only the tasks requiring massive memory from the four websites, excluding the cross-site setting, and evaluated performance with and without the scratchpad.
>
> |  | Wo Scratchpad | W Scratchpad |
> | --- | --- | --- |
> | Gemini2.5Pro | 35.5 | 38.5 |
> | GPT-5 | 50.3 | 47.9 |
>
> We observe that the results showed no significant impact.
> When examining the results in detail, we found that out of these 169 tasks, GPT-5 used the scratchpad in only 12 cases and Gemini in only 29 cases. A closer inspection showed that the scratchpad was not used effectively: the instruction sometimes caused the agents to check the scratchpad unnecessarily, or after checking it, to navigate to unrelated pages and ultimately fail the task. Because of these issues, the scratchpad did not provide meaningful benefits, and therefore, we did not include these results in the main experiments.
>
> > ###  **W3. It would be great to provide more details on how different modality is used for the agent’s action.**
>
> Thank you for pointing it out.  Our tasks are categorized as follows: Solvable with any observation (451 tasks), Solvable with the A11y Tree (69 tasks), and Solvable with a screenshot (12 tasks). Each task’s configuration file clearly specifies this information in the `required_obs` field.
>
> Each category uses the corresponding modality input as described below.
>
> |  | Solvable with any obs (451) | Solvable with A11y (69) | Solvable with Screenshot (12) |
> | --- | --- | --- | --- |
> | Input Modality | A11tree | A11tree | ScreenShot + A11tree |

---

### Official Review · Reviewer_cMr3 · 2025-11-05

**Soundness:** 3
**Presentation:** 3
**Contribution:** 3
**Rating:** 6
**Confidence:** 3

**Summary:**

The paper proposes a new benchmark called WebChoreArena, built on top of the WebArena simulation environments, WebChoreArena ensures strict reproducibility and enables fair, direct comparisons with the established WebArena benchmark, offering key insights into agent progress. This new benchmark comprises of 532 carefully curated tasks designed to extend the scope of WebArena beyond general browsing to more labor-intensive and tedious tasks. WebChoreArena systematically integrates three key challenges: 1) massive memory, 2) calculation, and 3) long-term memory. The experimental results demonstrate that as LLMs evolve, represented by GPT-4o, Claude 3.7 Sonnet, and Gemini 2.5 Pro, significant improvements in performance are observed on WebChoreArena. These findings suggest that WebChoreArena is well-suited to measure the advancement of state-of-the-art LLMs with greater clarity. Nevertheless, the results also indicate that even with Gemini 2.5 Pro, there remains substantial room for improvement compared to WebArena, highlighting the increased challenges posed by WebChoreArena.

**Strengths:**

Benchmark is well established on an existing infrastructure by WebArena, and following suit the best practices also makes this benchmark very reproducible and realistic, including good outcome-based reward to prevent reward hacking.

Code repository and containers are well organized online and reproducible, documentation is good

The paper is clear and reasonable about why this benchmark is needed, on top of the existing WebArena benchmark. It's focus on chore tasks, with longer horizon but more repetitiveness is indeed valid concern and justifies the new benchmark to differentiate from existing. The large memory required is quite interesting indeed and can test the model's capability.

**Weaknesses:**

As a benchmark paper, it would always be nice to present a bigger leaderboard and allow people to submit results. It mainly uses AgentOccam and BrowserGym as the agent framework, with several strong base models, but it would be really adding strength with more.

For Memory-intensive long horizon tasks, it would be nice to experiment with agent orchestration that has special focus on compression/condensation of observations, instead of more generic ones. One example could be OpenHands with has condenser feature.
More details about the data distribution and what is done to improve the data annotation matches what real world scenario presents is crucial. Given the environments are simulated, it's even more important to have realistic queries/tasks that closes the sim2real gap.

**Questions:**

See above

---

> ### Author Response · Authors · 2025-11-22
> **Rebuttal**
>
> We sincerely appreciate that the reviewer has taken the time to thoroughly understand our work and to highlight several strengths, including the importance of building on the well-established WebArena environments to ensure reproducibility and realism, the practical focus on chore-like tasks, and the clear differentiation from existing benchmarks.
>
> We have carefully considered your suggestions and updated the manuscript accordingly, with changes highlighted in magenta. Below, we provide detailed responses to each of your points:
>
> *Note: Unless otherwise specified, the mentioned line numbers denote those of the original submitted paper version instead of the rebuttal revision version.*
>
>
> > ### **W1. The creation of the leaderboard**
>
> Thank you for your advice. We will create a leaderboard and invite submissions of agent results from both our own experiments and the broader community.
>
> > ### **W2. Can you evaluate with e.g., OpenHands, which has a special focus on memory-intensive tasks?**
>
> Thanks for the suggestion. We additionally evaluated our benchmark using **OpenHands Cloud**, following the configuration recommended by OpenHands (Claude Sonnet 4). This setup reflects the performance of the most widely used OpenHands deployment.
>
> We ran experiments on the 102 subset tasks used in Table 2.
>
> The results are shown below.
>
> |  | Shopping | Admin | Reddit | GitLab | Average* |
> | --- | --- | --- | --- | --- | --- |
> | OpenHands | 28.0 | -* | 35.0 | 46.4 | 36.5 |
> | BrowserGym  | 40.0 | 51.7 | 15.0 | 42.9 | 34.0 |
>
> **For the Shopping Admin (Magento Admin Panel), OpenHands Cloud encountered webpage interaction errors on certain pages that do not occur for humans or other agents, so we omitted the results.*
>
> From these results, we observe that while using agents with condenser features, such as OpenHands, is indeed a promising direction for WebChoreArena, even this agent still leaves substantial room for improvement on our benchmark.
>
>
> > ### **W3.1. Did you design natural queries and tasks that reflect real user behavior?**
>
> While real-world evaluation is certainly important, we believe that evaluation in simulation environments offers certain advantages in reproducibility and sustainable evaluation that real-world evaluation cannot provide. Within this simulation setting, we instructed annotators to create queries that reflect plausible real-world use cases, rather than simply making the tasks artificially difficult.
>
> We prioritized constructing the tasks in simulation environments to ensure accurate and sustainable measurement of progress.
> Among simulation environments, the WebArena environment faithfully reproduces real-world webpages such as shopping sites, admin panels, GitLab, and Reddit, and therefore inherently exhibits a relatively small sim-to-real gap. As a result, the findings in this study should generalize well to real-world settings.
>
>
> > ### **W3.2. Evaluating the real-world performance gap**
>
> We agree that it is also important to carefully verify the gap between the real world and the simulation environment. To address this, we evaluated the performance gap in real-world settings.
>
> We constructed a total of 70 real-world chore tasks and measured agent performance on them.
>
> |  | WebChoreArena | Real |
> | --- | --- | --- |
> | GPT-4o | 3.4 | 5.7 |
> | Gemini2.5-Pro | 36.8 | 24.3 |
> | GPT-5 | 48.3 | 31.4 |
>
> As a result, we found performance trends that closely mirrored those in WebChoreArena (GPT-4o < Gemini 2.5-Pro < GPT-5).
>
> However, we also found that real-world websites change their content frequently, which makes it hard to verify results, and that many websites block agent actions for security reasons. Because of these issues, it is difficult to measure progress in a stable and sustainable way.
>
> Therefore, we argue that a benchmark like WebChoreArena is highly valuable, as it allows accurate evaluation within simulation environments.

---

### Author Response · Authors · 2025-11-22
**General Response on the Updates to the Paper**

We thank all the reviewers for their valuable effort, feedback, and comments. We are happy to hear that reviewers recognized our paper's strengths: benchmark’s focus on an important problem (cMr3, 2m48, HpZT), its full reproducibility (cMr3, HpZT), the careful curation process (2m48, 5TKV), and the overall solid benchmarking effort (5TKV, HpZT).



In response to the reviewers’ feedback, we have revised the corresponding sections of the paper. All modifications are highlighted in magenta.

**Reviewer cMr3**

We added the following description and results:

- Additional experiments with agents and models (Sec. D.2 and Table C)
- Experiments in real-world settings (Sec. D.5 and Table F)

**Reviewer 2m48**

We added the following description and results:

- More description of the analysis when using the calculator (Sec. D.7)
- Experimental results using a notebook (Sec. D.6, Table G)
- Detailed description of the modality input (Sec. 3.3, Sec. C.2, Table A)

**Reviewer 5TKV**

We added the following description and results:

- Checkpoint evaluation (Sec. D.1 and Table B)

**Reviewer HpZT**

We added the following description and results:

- Experiments with open-source models (Sec. D.3 and Table D)
- Analysis of reasoning effort (Sec. D.4 and Table E)
- Experiments with additional agents and models (Sec. D.2 and Table C)
- Experiments in real-world settings (Sec. D.5 and Table F)

---

### Author Response · Authors · 2025-12-02
**Summary of Reviews and Rebuttal**

Dear AC,

We would like to express our deep appreciation for the AC’s dedicated and tremendous efforts under these extraordinary circumstances.

This message provides a concise summary of the reviews we received and the key points of our rebuttal.

-----------------
### **1. Summary of Initial Review**

All reviewers assigned “3: good” in Soundness, Presentation, and Contribution, indicating overall satisfaction with the quality and significance of the paper. They recognized the significance of our work: benchmark’s focus on an important problem (**cMr3, 2m48, HpZT**), its full reproducibility (**cMr3, HpZT**), the careful curation process (**2m48, 5TKV**), and the overall solid benchmarking effort (**5TKV, HpZT**).

-----------------
### **2. Summary of Rebuttal**

As summarized below, reviewers requested additional experiments, all of which we conducted and fully addressed, including updating the draft accordingly. Although some concerns were raised about the project scope, we provided careful and thorough rebuttals. Several reviewers also asked for a detailed description, and we incorporated such feedback with revisions.

**Additional experiment**

- **cMr3, Additional OpenHands experiments:** We added the requested OpenHands results and showed that there remains substantial room for improvement even when using OpenHands (Sec. D.2 and Table C).
- **2m48, Notebook experiments:** We added preliminary notebook-based results (Sec. D.6, Table G) and showed that WebChoreArena can serve as a foundation for future efforts toward developing more advanced notebook capabilities.
- **5TKV, Checkpoint evaluation:** We clarified our support for checkpoint evaluation via checkpoint_url and added corresponding experiments (Sec. D.1 and Table B).
- **HpZT, Open-source models, reasoning depth, and additional agents/models:**  We added these experiments and the corresponding analysis (Sec. D.2, D.3, D. 4, and Table C, D, E).

**Project scope**

- **5TKV, Importance of chore tasks:** Regarding the importance of the chore tasks, only Reviewer 5TKV questioned their significance. However, we cited a recent large-scale survey of workers showing that the automation of tedious tasks is becoming increasingly important. Furthermore, Reviewer cMr3 and Reviewer 2m48 explicitly affirmed the importance of these tasks. Therefore, we believe that the significance of the chore tasks is well justified.
- **HpZT, Benchmark-focused contribution:**  We argued that the construction of a high-quality benchmark and its thorough analysis constitute a substantial contribution to the Datasets and Benchmarks area at ICLR, with examples from ICLR 2025.
- **HpZT, Manual construction:**  We argued that many well-established benchmarks also rely heavily on manual construction, and that manual annotation is essential for ensuring the quality, reliability, and robustness of the benchmark.
- **cMr3, HpZT, Sim2Real gap:** Two reviewers acknowledged the importance of conducting reproducible evaluations in simulation environments like WebChoreArena, yet they also expressed concerns about the sim-to-real gap. We re-emphasized the advantages of simulated evaluation and also provided real-world experiments showing a small sim-to-real gap.

**Clarification**

- **2m48**, **Detailed description of modality input:** We provided a more detailed explanation (Sec. 3.3, Sec. C.2, Table A).
- **2m48, More description of calculator result:** We added more description of calculator results (Sec. D.7).
- **HpZT,** **Suggestions for improvement:** We clarified that the benchmark highlights actionable weaknesses in current agents.
- **5TKV**, **Agent capability:** We argued that the agents evaluated in our work are strong open-source agents capable of fully leveraging the underlying model. We also noted that WebArena has already been used to evaluate closed-source agents behind systems like ChatGPT, and that its saturation motivates the need for WebChoreArena.

-----------------
We greatly appreciate the time, thoughtful consideration, and considerable effort the AC has devoted to managing this challenging review process.

Sincerely,
The Authors

---

### Meta-Review · Area_Chair_vbyW · 2026-01-07

**Summary:**

Overall, the authors made a great effort to address reviewer concerns through additional experiments and clarifications. They added new baselines (including OpenHands), explored notebook usage, and reported extra evaluations on real-world tasks. Some reviewer concerns were adequately addressed, particularly those related to baseline coverage and agent design choices. However, concerns about the sim-to-real gap remain only partially resolved. In particular, while real-world results were reported, the task collection and curation process was not clearly specified. The cited reference from Shao et al. does not need to fully justify the reality of the tasks collected in this benchmark.

**Reviewer Concerns:**

Addressed:

1. Baselines (OpenHands)
2. Scratchpad use
3. Additional ablations (e.g., use of different input modalities)

Outstanding:

Sim-to-real gap.

**Reviewer Scores:**

Reviewer cMr3 asked about the OpenHands model and the real-world performance gap. The authors provided additional evaluations on real-world tasks. However, I did not see how exactly those tasks were collected. I do not anticipate they will change their score based on the current result.
Reviewer 2m48 asked about additional ablations (e.g., calculator use, notebook use, different input modalities). The authors addressed all of these in their rebuttal. This reviewer should increase their score.
Reviewer 5TKV's main concern is about the sim-to-real gap, as the tasks in the benchmark were not collected in real-world use cases. The authors cited one reference, but do not seem to fully address this question. I do not anticipate they will change their score based on the current result.
Reviewer HpZT asked about additional baselines and LLM models, as well as real-world usage. The authors have addressed their concern by providing additional clarifications and results.

---

### Decision · Program_Chairs · 2026-01-26

Reject